# Vibroscape analysis reveals acoustic niche overlap and plastic alteration of vibratory courtship signals in ground-dwelling wolf spiders

Noori Choi [1,3] ✉, Pat Miller[2] & Eileen A. Hebets[1]

To expand the scope of soundscape ecology to encompass substrate-borne vibrations (i.e. vibroscapes), we analyzed the vibroscape of a deciduous forest floor using contact microphone arrays followed by automated processing of large audio datasets. We then focused on vibratory signaling of ground-dwelling *Schizocosa* wolf spiders to test for (i) acoustic niche partitioning and (ii) plastic behavioral responses that might reduce the risk of signal interference from substrate-borne noise and conspecific/heterospecific signaling. Two closely related species - *S. stridulans* and *S. uetzi* - showed high acoustic niche overlap across space, time, and dominant frequency. Both species show plastic behavioral responses - *S. uetzi* males shorten their courtship in higher abundance of substrate-borne noise, *S. stridulans* males increased the duration of their vibratory courtship signals in a higher abundance of conspecific signals, and *S. stridulans* males decreased vibratory signal complexity in a higher abundance of *S. uetzi* signals.

[1] School of Biological Sciences, University of Nebraska-Lincoln, Lincoln, NE, USA. [2] University of Mississippi field station associate, Abbeville, MS, USA. [3] Present address: Max Planck Institute of Animal Behavior, Konstanz, Germany. ✉email: noori0824@gmail.com

Understanding the evolution and function of animal communication requires knowledge of when and where animals communicate in their natural environments, which can be extremely difficult to acquire in many species. Even if direct observations of signaling behavior in the field are successful[1,2], classical surveillance methods are limited in terms of both quality and quantity of data. These limitations are due to the challenges associated with monitoring and tracking numerous signals from multiple individuals simultaneously, often within a large spatial/temporal range[1–4]. These challenges become particularly difficult to overcome when communication occurs in sensory modalities beyond the range of human perception that cannot be directly detected by researchers (e.g., near-field sound or substrate-borne vibrations). In such instances, scientists are often forced to infer information regarding animal communication from indirect evidence such as habitat use or activity patterns of study species.

Major advances in our understanding of airborne acoustic communication have been made possible by technological advances in data collection that have ultimately led to a new subfield of study—soundscape ecology[5]. Soundscape ecology leverages advances in sound recording technology to collect and quantify biological, geophysical, and anthropogenic sounds over a large spatial and temporal range in natural communities[4–9]. Thus far, advances in soundscape ecology have predominantly focused on the collection and classification of well-documented airborne sounds[9–12] (but see [8] for freshwater soundscapes), and have overlooked the more common substrate-borne acoustical environment. Such a narrow focus greatly constrains our understanding of soundscape ecology, as it ignores the most diverse and representative species in a community. For instance, many arthropods, a major taxonomic group of most ecosystems, predominantly communicate using substrate-borne vibrations[13–16]. Such signaling is unfortunately not captured by most airborne sound sensors.

Recent studies using a portable laser Doppler vibrometer have demonstrated that soundscape ecology can be extended to explore 'vibroscapes' and 'ecotremology'[14–18]. Unfortunately, the limited detection range of a single device hinders investigations of vibroscapes in the field as it is particularly difficult to conduct studies across large temporal and spatial scales[15,18,19]. While the detection range can be improved by using multiple devices, it currently limits participation in the study of vibroscapes and ecotremology to laboratories and investigators with access to such expensive equipment. This arguably impedes our broader understanding of how ecological and environmental interactions with biological, geophysical, and anthropogenic vibrations in natural habitats influence vibratory communications and their functions. Such an understanding is critical to our understanding of animal communication, however, as these interactions could promote evolutionary changes in the spatial, temporal, and acoustical properties of communications of many species involved in the vibroscapes.

The technical advances more widely available in soundscape ecology studies, such as inexpensive airborne sound recording equipment and algorithms to classify airborne sounds, enable not only a better understanding of when and where animals communicate[1,3] but also how co-occurring species partition their acoustic niche in a local community[5,20]. An acoustic niche is a hypothetical construct, similar to niche space in niche theory[21,22], describing the variation in signaling behavior with multiple dimensions of space, time, and structural characteristics (e.g. dominant frequency). Natural overlap in an acoustic niche among co-occurring species may induce negative impacts on the efficiency of communication due to the potential risk of signal interference[23–26]. In particular, the acoustic niche overlap for sexual communication can cause reproductive interference that can lead to fitness costs including wasted time, energy, or gametes, and potential detrimental hybridization[24].

Due to the numerous costs of acoustic niche overlap, co-occurring species are expected to partition the local acoustic niche by using different signaling locations (space) or time windows for signaling[27–32] and/or diverging signal properties such as spectral ranges or temporal patterns[21,32–35]. Characterizing acoustic niche overlap in natural communities is imperative to understanding its putative role in signal diversification, microhabitat use, and behavioral plasticity among animal signalers.

New advances in community-wide airborne soundscape ecology have enabled the characterization of acoustic niche overlap in natural communities and demonstrated that similarities and/or differences in spatial/temporal acoustic signaling dimensions can be quantitatively measured[21,27,36–38]. In particular, as compared to the traditional survey by manual observation, soundscape analysis using autonomous sensors enables researchers to use detailed information about variations in local soundscapes in various spatial and temporal scales to investigate how animals respond to short-term or long-term exposure to abiotic and biotic noise and acoustic niche overlap[3,8,39–42]. These data, in turn, can be used to directly test our understanding of ecological and evolutionary processes, such as the effects of abiotic or biotic noise disrupting the acoustic niche partitioning[43] and/or behavioral plasticity[44,45].

Prior research suggests that the animals using substrate-borne vibratory signals may alter various signal characteristics to avoid acoustic niche overlap with co-occurring species in their natural habitat[46–49]. However, these prior studies rely on patterns of the geographic distribution of allopatric/sympatric populations[46–48], or on the variation in the types of host plants[49], to infer whether and how vibratory signaling animals partition their acoustic niche. To date, there have been only a few attempts to use vibroscapes to explore signaling behavior, including potential acoustic niche partitioning and/or overlap in space, time, or structure, among sympatric and closely related species. Šturm et al. analyzed the seasonal and diurnal variation in substrate-borne vibrations produced by insects living in hay meadows using field recording by a laser vibrometer[18]. Their findings suggested that species using vibratory communication partition the acoustic niche by seasonal and fine temporal variation in signaling behaviors. The authors notably discuss limitations of their study that are related specifically to costly equipment necessary for recording substrate-borne vibration (e.g. laser Doppler vibrometer) and practical difficulties for analyzing large audio datasets.

In the present study, we had three objectives. First, (Obj 1) we characterized a natural vibroscape through the development of (i) a technique to collect substrate-borne field vibrations across large spatial/temporal scales and (ii) automated sound filtering and detection of sound events in large audio datasets. Second, using these techniques, we (Obj II) investigated how three focal sympatric species of *Schizocosa* wolf spider (*Schizocosa duplex* Chamberlin 1925, *S. stridulans* Stratton 1991, *S. uetzi* Stratton 1997) partition the spatial, temporal, and spectral dimensions of the acoustic niche within the vibroscape. Finally, we explored whether and how behavioral plasticity in the courtship signaling behavior of *Schizocosa* wolf spiders is shaped by (Obj IIIa) the noisiness of the local vibroscapes by general abiotic/biotic substrate-borne vibrations and (Obj IIIb) conspecific/heterospecific vibratory signals between two sister species – *S. stridulans* and *S. uetzi*.

## Methods

**Objective I—Characterize deciduous floor vibroscapes.** For field recordings, we chose five study plots (10 m × 10 m) at the field station of the University of Mississippi at Abbeville, Mississippi, USA (34°43′ N 89°39′ W). Before we chose study plot locations, we checked for the presence of mature and immature ground-dwelling wolf spiders by direct observation (Fig. 1a). To encompass the

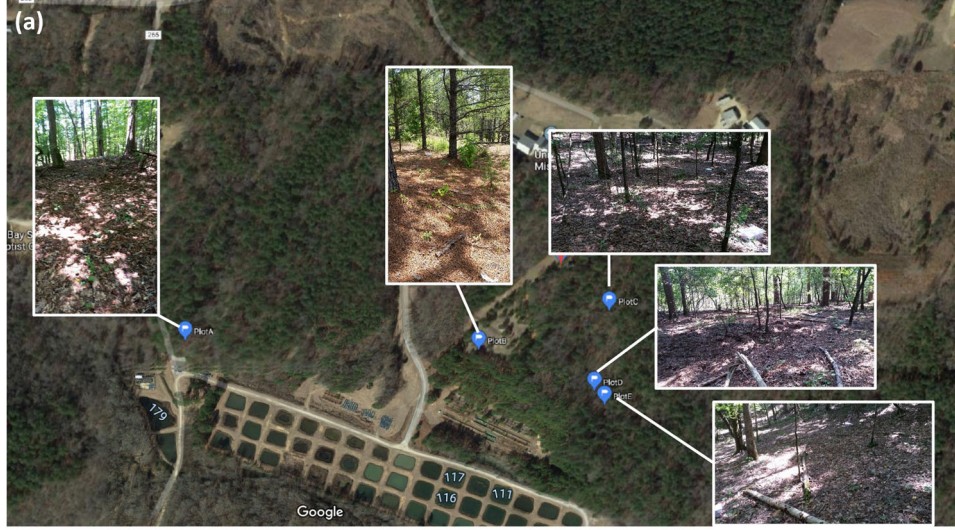

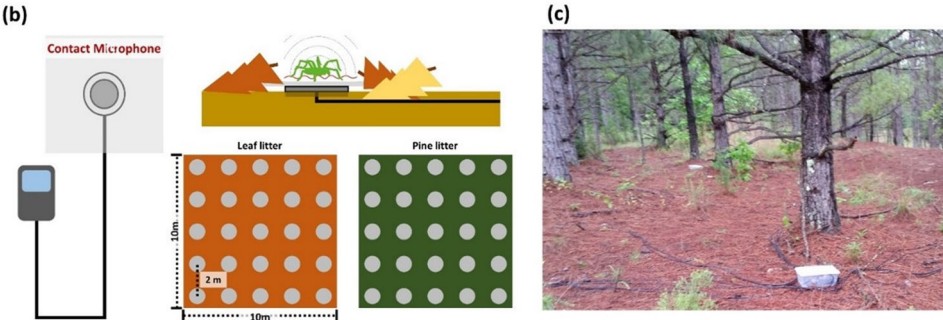

**Fig. 1 Study sites and experimental design for the field recording. a** Map of the study site and recording plots at the field station of the University of Mississippi. **b** Schematic diagram of the contact microphone array. **c** Field recording setup in one of the recording plots (plot B).

variation in species composition associated with substrate types, we chose to focus on two distinct microhabitats – (i) leaf litter and (ii) pine litter. Five study plots were covered by (i) leaf litter ($n = 3$ plots) and (ii) pine litter ($n = 2$ plots). In each study plot, we deployed 25 recording units consisting of a contact microphone (35 mm diameter, Goedrum Co., Chanhua, Taiwan) and a Toobom R01 8GB acoustic recorder (Toobom, China), a TemLog20 temperature logger (Tamtop, Milpitas, California, USA), and four pitfall traps (Carolina biological supply company, Bunington, North Carolina, USA) (Fig. 1b, c). In total, we deployed 125 recording units, 5 temperature loggers, and 20 pitfall traps across our five study plots. The temperature loggers recorded the temperature at each recording plot every 15 min during the experimental periods.

We placed recording units two meters apart for spatial independence of recorded substrate-borne vibrations and we adhered a circular waterproof paper (0.2 m radius) to a contact microphone to extend the detection range (Fig. 1b). We chose two meters because this is beyond the distance that substrate-borne vibratory signals of *Schizocosa* wolf spiders are known to travel[50], thus reducing the possibility that multiple recorders would be simultaneously picking up a single individual. Thus, each recording unit could be analyzed as an independent data set. We used propylene glycol for pitfall traps to minimize the potential environmental toxicity[51].

We conducted a 24-h recording every three days from May 15th to July 15th, 2018 resulting in a total of 1950 24-h recordings across 13 days. The substrate-borne vibrations during 24 h in study plots were continuously recorded from 0800 except 10 min to replace audio recorders at 1600 due to the limited battery capacity. To

compensate for the variation in the starting time of recording by travel time among study plots (~45 min between the first and the last plots), we conducted the recordings for at least 25 h to analyze the same 24-h period across the recording plots. After a ~24-h recording, we extracted uncompressed WAV files at a 48-kHz sampling rate from recorders to an 8 TB external hard drive (Seagate Technology LLC., Cupertino, California, USA). On the same day, we collected specimens from pitfall traps at three different times (0800, 1600, and 0000) to observe the temporal variation in the activity of species in study plots. We sorted collected specimens by the time of collection, collection date, and study plot and we preserved them in 95% ethanol for later species identification by PM. We used the collected specimens to corroborate our species identity of sound recordings across locations.

To automate signal detection for classification across our 125 recording units, we wrote Python programs to filter background noise, detect pulses, and group pulses into signal bouts. Before the process, we divided each 24-h recording WAV file into 10-min chunks using FFmpeg[52] for processing speed. We used the Crane cluster of Holland Computing Center at the University of Nebraska-Lincoln. We verified the automated methods for signal detection and noise filtering in Supplementary Material S1 (Supplementary Table S2; Supplementary Figs S2–S5).

Due to the spatial/temporal variation in background noise, we conducted adaptive noise filtering using a unique frequency spectrum of the background noise of each 10-min WAV chunk. To acquire the frequency spectrum, the program calculated the amplitude threshold. The amplitude threshold is calculated by sigma clipping as $m + \alpha\sigma$ with median $m$ and a standard

deviation σ of the amplitudes (mV) of all the sampling points of the WAV chunk. The constant α was determined among values between 1 to 10 at intervals of 0.3 by the elbow method on the number of sampling points above the amplitude threshold. Thus, depending on the background noise level, each WAV chunk has different amplitude thresholds for finding the longest silence to calculate the frequency spectrum for noise filtering. Once the amplitude threshold was determined, the program extracted the frequency spectrum of the longest segment below the amplitude threshold by Fast Fourier Transformation and filtered the WAV chunk by the frequency spectrum of background noise. The program for adaptive noise filtering was written based on Python packages including the *detect_silence* function of Pydub[53], kneed[54], and Noisereduce[55]. A detailed explanation of the methods is in Supplementary Data S1.

After noise filtering, we updated the amplitude threshold of each file by the sigma clipping methods through the same methods with noise filtering to find the optimal alpha. Using the updated amplitude threshold, we detected pulses above the amplitude threshold. The amplitude and time of detected pulses within a WAV chunk were recorded for pulse grouping and sound classification. For pulse grouping, the program calculated the time interval between adjacent detected pulses and applied the Gaussian Mixture Model (GMM) to classify the time intervals into three categories of within-bout (i.e. pulses detected from continuous production of sound without a pause), between-bout within a single signaling activity, and between-signaling activities. Then, we grouped pulses into bouts by the results of the GMM for sound classification. The program for pulse detection/grouping was written based on Python packages including the *find_peaks* function of Scipy[56] and the *GaussianMixture* function of the Scikit-learn package[57]. A detailed explanation of the methods and codes are in Supplementary Data S1 and Supplementary Software.

An expert in spider sound analysis (NC) classified detected sounds by visual inspection of spectrograms. To classify non-spider sounds, we used BirdNET[58], the Library of Singing Insects of North America (SINA)[59], and field observation (e.g. noise from airplanes). For the BirdNET, we accepted the species that showed the highest probability values from the online bird sound identification system. If the intervals between consecutive conspecific (or same class) sounds were recorded at the same vibratory sensor within one minute, we grouped the sounds as a signal bout. Also, if conspecific sounds from the same recording plot were detected by multiple sensors simultaneously, we classified the sounds as airborne sounds that were transmitted to the ground and counted the bouts as a single signal bout. When we cannot specify a reliable species or source producing sounds, we label the sound types as unknown (see Supplementary Audio).

**Objective II—Quantify acoustic niche partitioning among three Schizocosa species.** Within our vibroscape recordings, we focused on three co-occurring species of *Schizocosa* wolf spider – *S. duplex*, *S. stridulans*, and *S. uetzi*. Male *S. duplex* produces a stationary vibratory courtship signal with no visual component and this species is found mostly on pine litter[60,61]. Male *S. stridulans* produce multimodal courtship displays consisting of two discrete substrate-borne vibratory components—revs and idle[62] and visual signal components including foreleg ornamentations and leg tapping[63]. Male *S. uetzi* also produces stationary vibratory courtship signals that involve static/dynamic visual signal components including ornamentation on the forelegs and leg-arching behavior[64]. Among the three species, *S. stridulans* and *S. uetzi* are proposed sister species[65,66] and are often found in the same leaf litter habitat[64]. *Schizocosa duplex* is the next closest relative to the *S. stridulans* - *S. uetzi* species pair[65].

Based on the information about the space (i.e. substrate type of recording plots), time (i.e. date and time of detected signal bouts), and spectral range (i.e. dominant frequency range), we quantified the interspecific acoustic niche overlap between different sound types included in our three focal species using Pianka's niche overlap index (PI, 0 – No overlap, 1 – complete overlap)[67]. Pianka's niche overlap index is computed by:

$$PI = \left(\sum_{n=1}^{k} P_{ik}P_{jk}\right) \Big/ \sqrt{\sum_{n=1}^{k} P_{ik}^2 \times \sum_{n=1}^{k} P_{jk}^2} \qquad (1)$$

where $P_{ik}$ is the proportion of $i$th resource (e.g. space – leaf litter & pine litter; time – time windows; spectral range – frequency range) of the resource used by species $i$. To quantify Pianka's niche overlap index, we divided each 24-h recording into 15-min time bins and dominant frequencies into 10 Hz-frequency bins. Then, we quantified Pianka's niche overlap index for signaling time by multiplying the indices for the date (seasonal variation) and time of a day (diurnal variation) of the detected signal bouts. We did not quantify Pianka's niche overlap index with airborne sounds because of the potential differences in niche dimensions (e.g. microhabitat—leaf litter vs. pine litter—for vibratory signaling).

**Objective III—Examine behavioral plasticity in the courtship signaling behavior of Schizocosa wolf spiders.** We tested whether there was significant variation in bout duration and dominant frequency of our focal *Schizocosa* species due to the abundance (i.e. total number of detections) and/or diversity (i.e. Shannon diversity index of detected sounds) of general noise (all detected airborne and substrate-borne vibrations other that belong to different types) (Obj IIIa). Following our discovered high acoustic niche overlap between *S. stridulans* and *S. uetzi*, we also tested whether there was significant variation in the same signal characteristics between these two species in relation to the abundance of conspecific/heterospecific vibratory signals (Obj IIIb). To investigate the realistic effects of general noise and conspecific/heterospecific vibratory signals that the animal experiences during the signal production, we measured the abundance and diversity of the general noise and conspecific/heterospecific signals during ± 15 min of each signal bout (hereafter, *interaction time window*) for each species.

In addition to the two signal characteristics mentioned previously (bout duration and dominant frequency), we also explored the effects of variation in the abundance and diversity of general noise (Obj.IIIa) and conspecific/heterospecific signals (Obj.IIIb) on the structural complexity of detected courtship signals of *S. stridulans*. We focused on complexity in *S. stridulans* specifically, as prior research has demonstrated an influence of male vibratory signal complexity on mating success[68]. Among the vibratory signal components (i.e. revs and idles), idles and the associated visual signal (i.e. foreleg tapping) mainly influence the complexity of male courtship signals[68]. Previous studies suggested that (i) males producing more complex vibratory signals are more likely to mate with females[68,69] and (ii) males can plastically alter the vibratory signal complexity according to female body mass by removing or adding idles in the courtship signal sequences[68]. Thus, we used the duration of idles in detected signal bouts as a proxy for the signal complexity of *S. stridulans*.

To investigate the effects of interspecific acoustic niche overlap on vibratory signals of three focal *Schizocosa* wolf spiders, we measured the duration and dominant frequency of detected signal bouts. We chose these two characters due to their relative robustness to our noise filtering method which may distort the measurement of other acoustic characters depending on the background noise profiles (e.g. frequency range, signal-to-noise ratio; Supplementary Table S2; Supplementary Figs. S1–S5). To

eliminate the effects of silence between actual signals within a bout on the quantification of dominant frequency, we used the median values of the dominant frequency of non-silence parts in a bout. We quantified dominant frequency with the Short-Time Fourier Transform (STFT) with a window length of 0.1 seconds and hop length of 0.05 seconds using the *pyin* function of the librosa Python package[70].

We quantified the diversity of general noise by calculating the Shannon entropy of the types of noise. The Shannon entropy (H)[71] is defined as:

$$H(X) = \sum_{i=1}^{n} P(x_i) \log P(x_i) \qquad (2)$$

where the proportion of a type of substrate vibration, $P(x_i)$, of $n$ substrate vibration types occurred during the interaction time window of each signal bout. We normalized the Shannon entropy by the maximum entropy, $\log(n)$, due to the variation in the number of substrate vibration types across signal bouts.

**Statistics and reproducibility**. To investigate the effects of the noisiness of the local vibroscape (Obj. IIIa) for each species, we tested the effects of the abundance and diversity of general noise that occurred during the interaction time window of each signal bout on the signal properties using mixed-effect linear regression. We used the abundance, diversity, and the interaction term of general noise as predictor variables and bout duration and dominant frequency as the response variables. We used temperature during signaling as a random effect.

To test the effects of abundance of conspecific/heterospecific signals between closely related species (*S. stridulans* vs. *S. uetzi*), we quantified the relative abundance of conspecific/heterospecific signals. To quantify the relative abundance, we divided the number of conspecific/heterospecific signals during the interaction time window of each signal bout by the maximum value across the whole recording period (*S. stridulans*: 7 times, *S. uetzi*: 11 times) so that the value ranges from 0 to 1. Then, we conducted mixed-effects linear regression with the relative abundances of conspecific and heterospecific signals and the interaction term as the predictor variable, temperature during signaling as random effects, and bout duration and dominant frequency as the response variables.

We investigated the effects of the abundance and diversity of general noise that occurred in the interaction time window of each signal bout on the duration of idles in an *S. stridulans* courtship signal bout through the zero-inflated mixed-effect Gamma regression. We used the abundance, diversity of general noise, and the interaction term as predictor variables and the duration of idle in a signal bout as the response variables. Also, we tested the effects of the relative abundance of conspecific/ heterospecific (*S. uetzi*) *Schizocosa* signals on the duration of idles in *S. stridulans* courtship signal bouts using the zero-inflated Gamma regression with the relative abundances of conspecific/ heterospecific signals and the interaction term as the predictor variable and the duration of idle in a signal bout as the response variables.

We used the lmer() and glmmTMB() functions in the lme4{}[72] and glmmTMB{} R package[73] for regression models. The p-values of predictors were calculated using the Anova() function of the car{} R package.

**Reporting summary**. Further information on research design is available in the Nature Portfolio Reporting Summary linked to this article.

## Results

**Objective I – Deciduous floor vibroscape**. Using 75 field recording units, we collected 17,713 signal bouts from 73 different types of substrate vibrations. We only used 17 sound types that occurred more than 100 times across recording periods for further analysis. The airborne substrate vibrations which are acoustic sounds transmitted to ground included numerous birds (American crow - *Corvus brachyrhynchos*, blue jay - *Cyanocitta cristata*, red-eyed vireo - *Vireo olivaceus*, pine warbler - *Setophaga pinus*, northern cardinal - *Cardinalis cardinalis*, yellow-breasted chat - *Icteria virens*, eastern whip-poor-will - *Antrostomus vociferus*, eastern wood pewee - *Contopus virens*), crickets (Jamaican field cricket - *Gryllus assimilis*), and airplane (Fig. 2). For substrate-borne vibrations, we identified three focal species of *Schizocosa* wolf spider – *S. duplex, S. stridulans*, and *S. uetzi*–in addition to additional unknown types of substrate vibrations (Fig. 2). The temporal variations of detected sounds in each plot can be found in Supplementary Material S1 (Supplementary Figs. S6– S10).

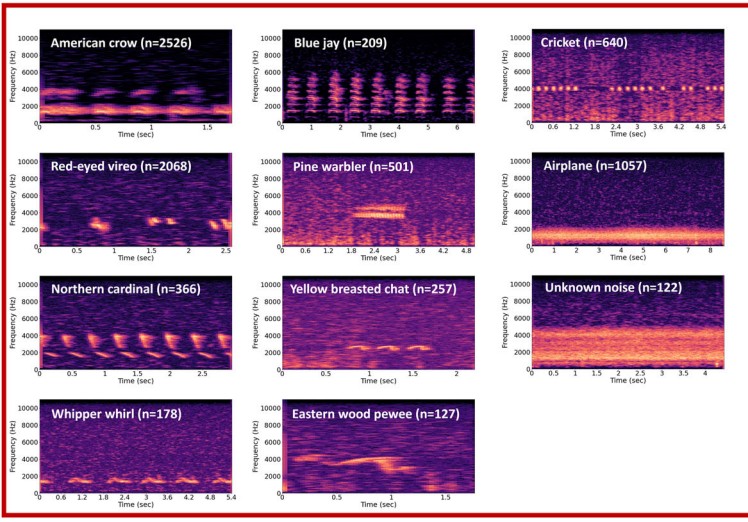 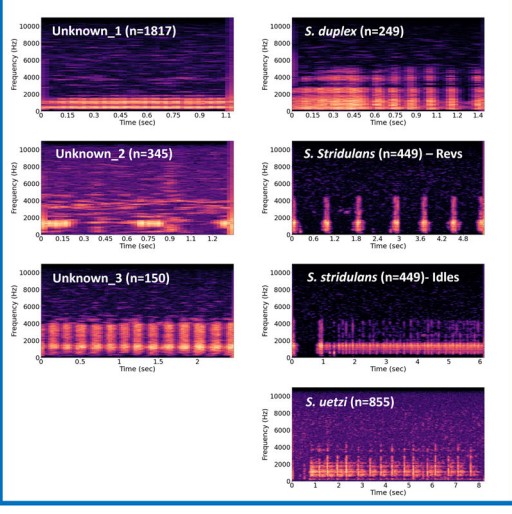

**Fig. 2 Spectrogram of sounds that were detected more than 100 times in the vibroscape.** Airborne sounds and substrate-borne sounds were grouped by different colors of boxes (Red—airborne sounds; Blue—substrate-borne sounds). The total number of detections was denoted in parentheses. *Schizocosa stridulans* produced two different types of vibratory signals; Revs and Idles.

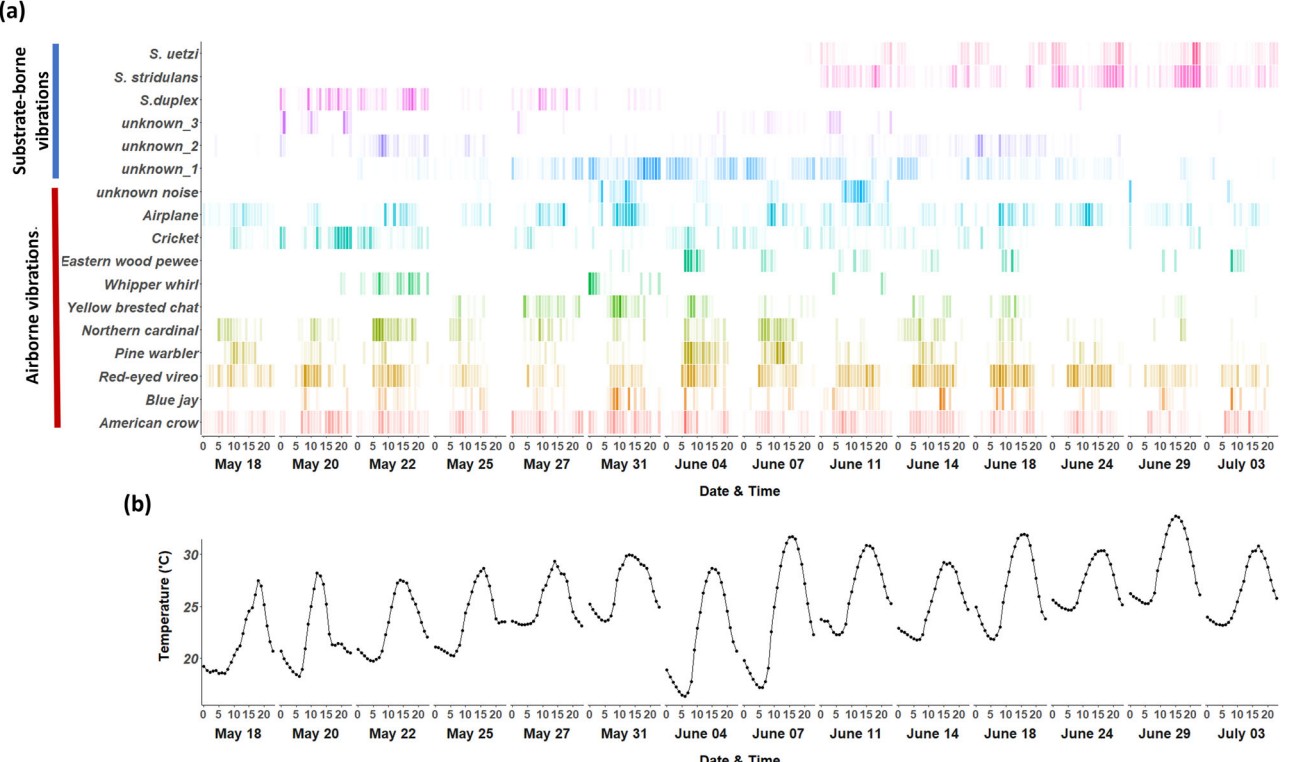

**Fig. 3 Integrated temporal distribution of airborne/substrate-borne vibrations from all recording plots with temperature data. a** Temporal distributions of airborne/substrate-borne substrate vibrations and **b** temperature during the experimental periods. The bin size for temporal distribution and temperature is 1 h. The number of detected bouts for each bin was represented by the gradient of corresponding colors. Sound types that were observed less than 100 times were excluded.

The developed techniques enabled us to acquire detailed information about (i) the spatial/temporal distribution of different types of airborne/substrate-borne vibrations in a local community (Fig. 3a, Table 1a) and (ii) environmental factors that potentially influence animal communication (i.e. temperature) (Fig. 3b). In particular, we can visualize the overlap in time across sounds and examine how the vibroscape varies with temperature through the deployment of temperature loggers (Fig. 3). We were also able to determine that *S. duplex* produces signals primarily on pine litter while *S. stridulans* and *S. uetzi* produce signals primarily on leaf litter (Table 1a).

The temporal variation in the number of specimens from pitfall traps showed similar phenological patterns of matured *Schizocosa* wolf spider males as the soundscape data (Fig. 4). Pitfall trapping resulted in the following total numbers of each species: *S. duplex*—19 males; *S. stridulans*—18 males; *S. uetzi*—24 males (Fig. 4a). We only identified males, as only males produce substrate-borne vibratory courtship songs, and the interspecific variation in the structure of female genitalia, which is the key to species identification, is indistinct between closely related species[66]. Typical field collections result in a nearly equal sex ratio (authors pers obs) suggesting that the total numbers of each species were likely double what we recorded, but we did not count the number of female *Schizocosa*. The temporal occurrence of vibratory signals of three *Schizocosa* wolf spider species was significantly correlated with the temporal variation in the number of collected male specimens from the pitfall trapping (Pearson correlation test; $r = 0.579$, $P < 0.001$, Fig. 4b).

**Objective II—Acoustic niche partitioning among three Schizocosa species.** Among three species of *Schizocosa* wolf spiders

**Table 1 (a) The number of collected bouts and (b-f) interspecific acoustic niche overlap among *Schizocosa* wolf spiders. Acoustic niche overlap was quantified by Piankas's niche overlap index. The maximum acoustic niche overlap was denoted by bold.**

|  | *S. duplex* | *S. stridulans* | *S. uetzi* |
|---|---|---|---|
| (a) The number of collected bouts |  |  |  |
| Pine litter | 247 | 2 | 0 |
| Leaf litter | 2 | 518 | 855 |
| Total | 249 | 520 | 855 |
| (b) Spatial overlap (Recording plot) |  |  |  |
| *S. duplex* | – | – | – |
| *S. stridulans* | 0.007 | – | – |
| *S. uetzi* | 0.006 | **0.922** | – |
| (c) Temporal overlap (Date) |  |  |  |
| *S. duplex* | – | – | – |
| *S. stridulans* | 0.038 | – | – |
| *S. uetzi* | 0.009 | **0.973** | – |
| (d) Temporal overlap (Time of day) |  |  |  |
| *S. duplex* | – | – | – |
| *S. stridulans* | **0.821** | – | – |
| *S. uetzi* | 0.562 | 0.750 | – |
| (e) Spectral overlap (Dominant frequency) |  |  |  |
| *S. duplex* | – | – | – |
| *S. stridulans* | 0.776 | – | – |
| *S. uetzi* | 0.754 | **0.891** | – |
| (f) Overall acoustic niche overlap |  |  |  |
| *S. duplex* | – | – | – |
| *S. stridulans* | <0.001 | – | – |
| *S. uetzi* | <0.001 | **0.599** | – |

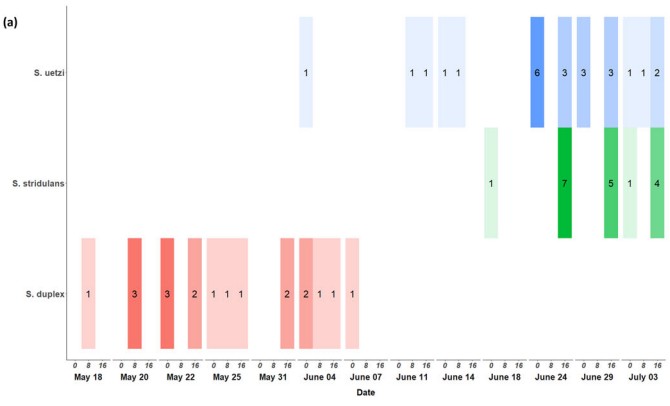
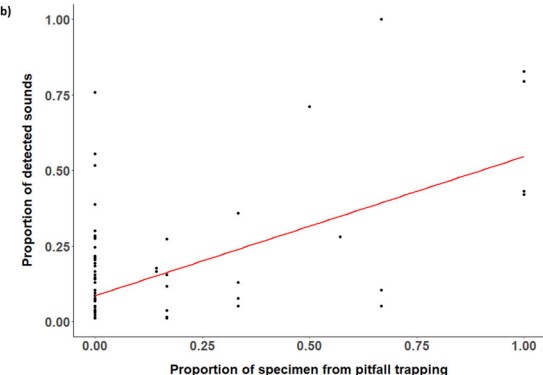

**Fig. 4 Comparison of the results of our vibroscape analysis with the traditional pitfall trapping method. a** Temporal distributions of collected *Schizocosa* males by pitfall trapping (*S. duplex*—orange; *S. stridulans*—green; *S. uetzi*—blue) and **b** the relationship between the number of collected specimens and detected vibratory courtship signals of *Schizocosa* wolf spiders. The number of collected specimens for each time bin was represented by the gradient of corresponding colors. The numbers on colored bars represent the number of collected *Schizocosa* males.

that were observed during the experimental period (*S. duplex, S. stridulans, S. uetzi*), *S. stridulans* and *S. uetzi* showed the highest acoustic niche overlap across space, time, and spectral properties of their vibratory signaling, with over 0.9 acoustic niche overlap in space and date, and over 0.75 in time of day (Table 1). *Schizocosa duplex* had a niche overlap index of 0.82 with *S. stridulans* and 0.56 with *S. uetzi* in time of day, but less than 0.1 with either species in spatial overlap or date overlap (Table 1).

The vibroscape data showed that *Schizocosa* wolf spiders encounter abiotic/biotic noise during their signaling that may potentially induce signal interference. At each recording unit, on average, 10.624 general noises (abiotic + biotic noise; maximum = 37, minimum = 1), 3.274 conspecific signals (maximum =12, minimum = 0), and 1.212 heterospecific signals (maximum = 11, minimum = 0) occurred before and after 15 min of *Schizocosa* wolf spider's signal bout.

**Objective III—Behavioral plasticity in the courtship signaling behavior of Schizocosa wolf spiders.** Across the three *Schizocosa* wolf spider species, the duration and dominant frequency of courtship signals were not significantly predicted by the abundance and diversity of general noise, except for the bout duration of *S. uetzi* (Fig. 5; Table 2a, b). *Schizocosa uetzi* males produced shorter courtship signals when they courted coincident with a high abundance of noise regardless of how diverse the noise was.

The duration of idles in signal bouts of *S. stridulans* was not significantly predicted by the abundance and diversity of noise, nor the interaction terms (Table 2c).

The bout duration of *S. stridulans* was significantly predicted by the abundance of conspecific signals and the interaction term, but the dominant frequency was not, nor was the interaction term. (Fig. 6; Table 3a, b). *Schizocosa stridulans* males produced longer courtship bouts in a higher abundance of conspecific signals, but there was no effect of the abundance of heterospecific signals. The bout duration and dominant frequency of *S. uetzi* were not significantly predicted by the abundance of conspecific/heterospecific signals or the interaction term (Fig. 6; Table 3a, b).

*Schizocosa stridulans* males were likely to decrease the duration of idles in the higher abundance of heterospecific signals but were not influenced by the abundance of conspecific signals. The interaction term was also a significant predictor of the duration of idles in a bout (Fig. 7; Table 3c).

## Discussion
We successfully characterized the vibroscape in a North American deciduous forest floor by recording both substrate-borne vibrations (e.g., spider courtship) and airborne sounds transmitted to the ground (e.g. bird and cricket song, anthropogenic noise—airplanes) using inexpensive contact microphone arrays. Also, through the successful automation of background noise filtering and sound detection, we extracted more than 10,000 bouts of sounds from multiple 24-h recording files (39,000 h of recording in total). From this large dataset of substrate-borne vibration recordings in the field, we were able to identify 10 airborne sounds and 4 substrate-borne sounds with 3 unknown substrate-borne sounds including courtship songs of three species of *Schizocosa* wolf spider.

Our recordings also aligned with our more traditional phenological assessment of species activity using collected specimens by pitfall trapping. Notably, the number of signals detected for each species far outnumber the specimens we collected in pitfall traps (Supplementary Table S1). The rich dataset of deciduous forest floor vibroscape allows us to quantify vibratory noise abundance, diversity, as well as species-specific signaling patterns of *Schizocosa* wolf spiders. Using these data, we were able to test hypotheses about naturally occurring acoustic niche overlap among closely related species and the behavioral responses of multimodal signaling wolf spiders (*Schizocosa duplex, S. stridulans*, and *S. uetzi*).

The vibroscape of our focal deciduous forest floors notably included both airborne and substrate-borne sounds, making this acoustical environment tremendously rich and noisy. Our recordings, which were accomplished through the use of inexpensive recording units (~ $10 per unit), highlight that animals communicating through substrate-borne vibrations are challenged with not only competition in space and time with other vibratory signaling animals but potentially with those using airborne songs and calls as well. This dual challenge of potential competition from substrate-borne and airborne sound has also been shown in previous studies[18,19,74]. The tendency of natural forest floor substrates to transmit airborne sounds as well as substrate-borne sounds is an underappreciated obstacle to the detection and interpretation of salient information for the tremendous biodiversity of animals living in this environment. Previous studies suggested that airborne sounds transmitted to the artificial substrates (e.g. filter paper) influence the substrate-borne communication of another ground-dwelling wolf spider in laboratory conditions (*Schizocosa ocreata*[74,75]), but further studies are required to understand whether and how the airborne sounds transmitted to natural substrates (e.g. leaf litter) influence the communication of ground-dwelling arthropods including spiders.

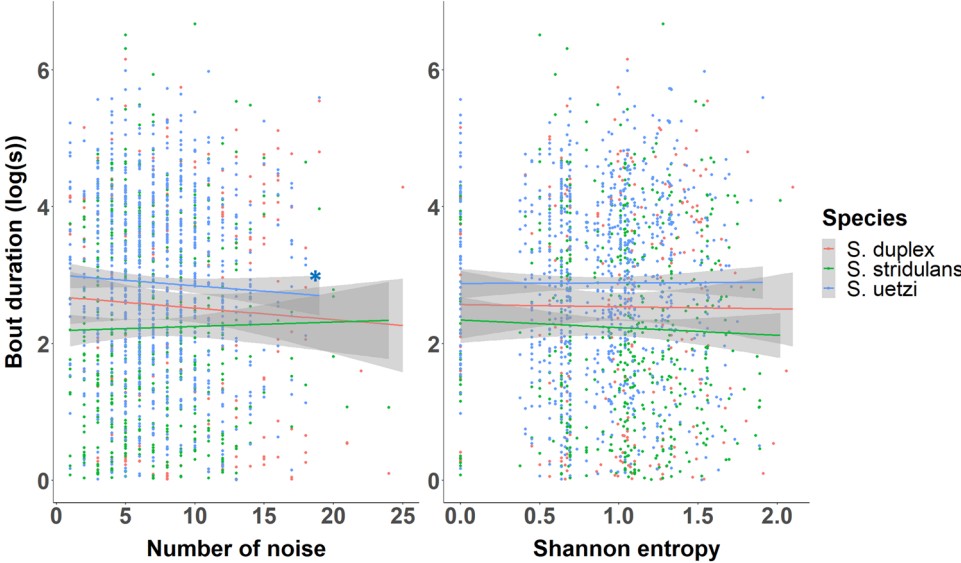

**Fig. 5 The variation in bout duration (second) of *Schizocosa* wolf spiders by the abundance and diversity of general noise during an interaction time window (±15 min from each courtship bout).** The diversity of general noise was quantified by the Shannon entropy of different types of noise. The species were color-coded (*S. duplex*–orange; *S. stridulans*—green; *S. uetzi*—blue) and the shaded bands denote 95% confidence intervals. The significant effects were denoted by asterisks with the same color code ($P < 0.001$ '***'; $P < 0.01$ '**'; $P < 0.05$ '*').

**Table 2 Results of mixed-effects linear regression of the effects of general noise during signaling behaviors on the signal properties of three species of *Schizocosa* wolf spiders. Significant results were denoted by bold.**

| Species | Abundance of noise | Shannon entropy | Interaction term |
|---|---|---|---|
| (a) Bout duration | | | |
| S. duplex (n = 249) | Wald $\chi^2_1 = 1.896$, $P = 0.169$ | Wald $\chi^2_1 = 0.000$, $P = 0.999$ | Wald $\chi^2_1 = 0.949$, $P = 0.330$ |
| S. stridulans (n = 520) | Wald $\chi^2_1 = 1.611$, $P = 0.204$ | Wald $\chi^2_1 = 0.047$, $P = 0.829$ | Wald $\chi^2_1 = 0.845$, $P = 0.358$ |
| S. uetzi (n = 855) | **Wald $\chi^2_1 = 4.547$, $P = 0.033$** | Wald $\chi^2_1 = 0.255$, $P = 0.613$ | Wald $\chi^2_1 = 2.327$, $P = 0.127$ |
| (b) Dominant frequency | | | |
| S. duplex (n = 249) | Wald $\chi^2_1 = 1.375$, $P = 0.241$ | Wald $\chi^2_1 = 0.001$, $P = 0.975$ | Wald $\chi^2_1 = 0.372$, $P = 0.542$ |
| S. stridulans (n = 520) | Wald $\chi^2_1 = 0.066$, $P = 0.798$ | Wald $\chi^2_1 = 0.000$, $P = 0.999$ | Wald $\chi^2_1 = 0.009$, $P = 0.925$ |
| S. uetzi (n = 855) | Wald $\chi^2_1 = 0.359$, $P = 0.549$ | Wald $\chi^2_1 = 0.002$, $P = 0.967$ | Wald $\chi^2_1 = 0.097$, $P = 0.756$ |
| (c) Duration of idles in signal bouts of S. stridulans | | | |
| S. stridulans (n = 520) | Wald $\chi^2_1 = 0.001$, $P = 0.974$ | Wald $\chi^2_1 = 0.286$, $P = 0.593$ | Wald $\chi^2_1 = 0.347$, $P = 0.555$ |

Based on the vibroscape analysis, we investigated the acoustic niche partitioning among *Schizocosa* wolf spiders. We found that vibroscapes appear to vary across seasons and time of day and that *S. duplex* is spatially isolated from *S. stridulans* and *S. uetzi* in signaling microhabitat use (pine litter vs. leaf litter respectively). *Schizocosa stridulans* and *S. uetzi* overlap in all dimensions of acoustic niche space (microhabitat use, seasonal/diurnal activity, and dominant frequency range) and did not show any evidence of acoustic niche partitioning—i.e. no evidence of reduced encounter rate in the spatial, temporal, and spectral acoustic niche dimensions. The spatial, temporal, and spectral overlaps between *S. stridulans* and *S. uetzi* (Pianka index: 0.750–0.922; Table 1b) are extremely high as compared to previous results about acoustic niche overlap of animals using airborne acoustic signals (Anuran, seasonal overlap: min. 0.21 – max. 0.62[76]; min. 0.11 – max. 0.34[77]; Avian, diurnal overlap: min. 0.167 – max. 0.547[33]; min. 0.059 – max. 0.844[78]).

The high spatial, temporal, and spectral overlap between *S. stridulans* and *S. uetz* suggests that there may be competition for the acoustic niche between these two sister species. While high acoustic niche overlap is not necessarily linked to signal interference, considering the importance of vibratory courtship signals

for species recognition in *Schizocosa* wolf spiders[79,80], the high acoustic niche overlap between two closely related *S. stridulans* and *S. uetzi* may induce fitness costs for both sexes such as wasted energy, time, or gametes[24]. In particular, signal interference between *S. stridulans* and *S. uetzi* may increase the risk of hybridization due to the similar structures of female genitalia[66] and the low preference for hybrid individuals of *Schizocosa* females (*S. ocreata* + *S. rovneri*)[79]. The potential impact of signal interference on the risk of hybridization between *S. stridulans* and *S. uetzi* should be tested with genetic studies across different populations that vary in the acoustic niche overlap between the two species.

The information from our vibroscape analysis provides insight into not only the acoustic niche overlap but also the potential mechanisms to avoid signal interference among closely related *Schizocosa* wolf spider species in our study site. Many empirical studies of signal interference in multi-species assemblages (reviewed in [24]), especially among related species sharing similar acoustic niches, demonstrate that avoidance of interference can be acquired by (i) reducing the encounter rate with neighboring species through changing space or time of signaling[24,81], (ii) minimizing the impacts of signal interference through the

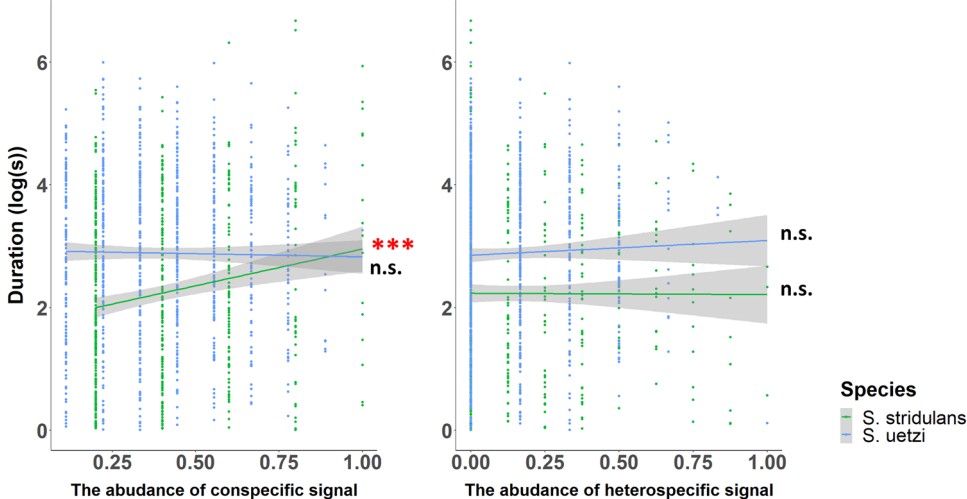

**Fig. 6 The variation in bout duration of *S. stidulans* and *S. uetzi* wolf spiders by the abundance of conspecific/heterospecific signals during the interaction time window of each signal bout.** The diversity of general noise was quantified by the Shannon entropy of different types of noiseThe species were color-coded (*S. stridulans*—green; *S. uetzi*—blue) and the shaded bands denote 95% confidence intervals. The significant effects were denoted by asterisks with the same color code ($P < 0.001$ '***'; $P < 0.01$ '**'; $P < 0.05$ '*').

**Table 3 Results of mixed-effects zero-inflated Gamma regression of the effects of general noise during signaling behaviors on the signal properties of three species of *Schizocosa* wolf spiders. Significant results were denoted by red shades.**

| Species | Abundance of conspecific | Abundance of heterospecific | Interaction term |
|---|---|---|---|
| (a) Bout duration | | | |
| *S. stridulans* ($n = 520$) | *Wald* $\chi^2{}_1 = 24.738$, $P < 0.001$ | *Wald* $\chi^2{}_1 = 4.319$, $P = 0.122$ | *Wald* $\chi^2{}_1 = 6.674$, $P = 0.015$ |
| *S. uetzi* ($n = 855$) | *Wald* $\chi^2{}_1 = 0.291$, $P = 0.590$ | *Wald* $\chi^2{}_1 = 0.081$, $P = 0.776$ | *Wald* $\chi^2{}_1 = 0.093$, $P = 0.760$ |
| (b) Dominant frequency | | | |
| *S. stridulans* ($n = 520$) | *Wald* $\chi^2{}_1 = 0.078$, $P = 0.780$ | *Wald* $\chi^2{}_1 = 0.579$, $P = 0.447$ | *Wald* $\chi^2{}_1 = 0.359$, $P = 0.549$ |
| *S. uetzi* ($n = 855$) | *Wald* $\chi^2{}_1 = 0.992$, $P = 0.319$ | *Wald* $\chi^2{}_1 = 1.674$, $P = 0.196$ | *Wald* $\chi^2{}_1 = 1.417$, $P = 0.234$ |
| (c) Duration of idles in signal bouts of *S. stridulans* | | | |
| *S. stridulans* ($n = 520$) | *Wald* $\chi^2{}_1 = 0.042$, $P = 0.839$ | *Wald* $\chi^2{}_1 = 5.080$, $P = 0.024$ | *Wald* $\chi^2{}_1 = 4.135$, $P = 0.042$ |

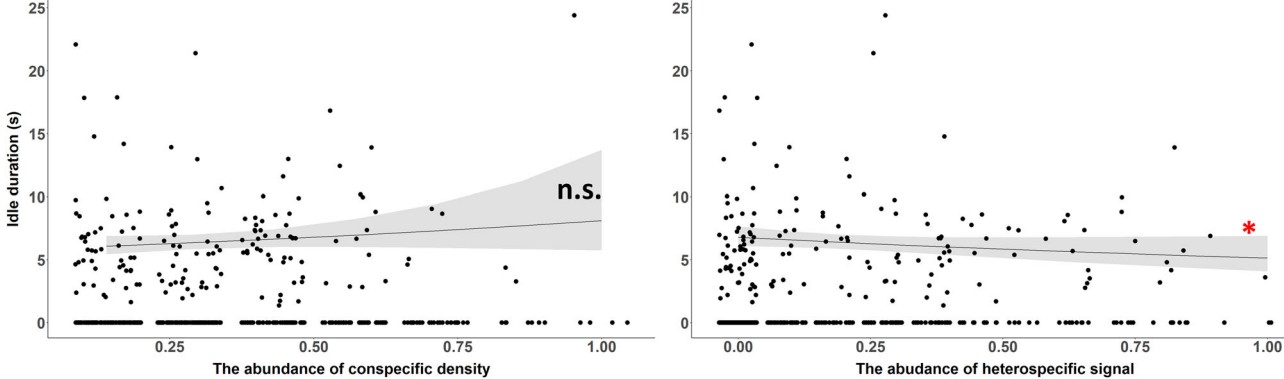

**Fig. 7 The predicted duration of idles in a signal bout of *S. stridulans* by the abundance of conspecific (Left) and heterospecific (Right) signals during the interaction time window of each signal bout.** The shaded bands denote 95% confidence intervals. The dots represent values from individual signal bouts of *S. stridulans*. The significant effects were denoted by asterisks with the same color code ($P < 0.001$ '***'; $P < 0.01$ '**'; $P < 0.05$ '*').

diversification of signal characteristics[43], (iii) short-term adjustment of signaling behavior such as spectral properties, amplitude, or timing when they encounter signal interference[82,83], or (iv) perceptual acuity of females for species recognition[23,84].

As highlighted previously, we found no evidence of (i) reducing encounter rates or (ii) diversifying signal characteristics as strategies for *S. stridulans* and *S. uetzi* to avoid signal interference.

Considering that the time, space, and spectral properties of vibratory signals of ectothermic arthropods are often highly constrained by physiological and environmental factors[13,85,86], the avoidance of signal interference by shifting space or time of signaling may not be selected for due to the detrimental effects on transmission efficiency on different substrates[87,88], male signaling behaviors[89], or female response to male courtship signals[90].

While potentially the divergence of acoustic characters can be explored more fully, our results suggested that *Schizocosa* wolf spiders are using (iii) short term adjustment of signaling behavior to reduce signal interference.

Our vibroscape analysis allowed us to investigate whether three focal *Schizocosa* wolf spiders might reduce the risk of signal interference by altering signal characteristics in response to temporal variation in the abundance and diversity of noise. While *S. duplex* and *S. stridulans* did not show a significant response to the variation in the abundance or diversity of noise, *S. uetzi* males did reduce the duration of their signal bouts when abiotic/biotic noise occurred more frequently (Fig. 6, Table 2). Similarly, a previous study suggested that male *Schizocosa* wolf spiders reduced their courtship activity in response to avian calls transmitted to the ground as a defense mechanism for the perceived predation risk (*S. ocreata*[75]). The reduced bout duration of *S. uetzi* associated with the abundance of general noise may also be induced by predator avoidance, but it is not clear why other species (*S. duplex* and *S. stridulans*) did not show a significant variation in their signaling activities.

*Schizocosa stridulans* males produced shorter courtship bouts when there were more signals produced by conspecific rivals (Fig. 7, Table 3). In other *Schizocosa* species (*S. retrorsa*), males also reduced the courtship duration responding to the increased conspecific male density[91]. This reduced courtship activity may suggest that *S. stridulans* males avoid producing courtship signals when male-male competition is perceived to be high. Testing this hypothesis would require observations of the actual number of males and females within the vibroscape, but unfortunately, we do not have that data.

*Schizocosa stridulans* also altered their signal structure in terms of the duration of specific components (i.e. idles) with the abundance of heterospecific *S. uetzi* signaling. In response to how abundant *S. uetzi* signals are around courting *S. stridulans* males, *S. stridulans* males decreased the complexity of their vibratory signals by reducing the duration of idles (Fig. 7, Table 3). This decreased signal complexity may be driven by the need for better species recognition. The structure of idle of *S. stridulans* is a continuous repetition of short pulses, which is similar to components of *S. uetzi* courtship signals. Thus, *S. stridulans* males may reduce the duration of idles to avoid potential misrecognition of species identity by females. This hypothesis may also explain why *S. stridulans* males reduced the bout duration in a higher abundance of *S. uetzi* signals (Fig. 7, Table 3). To test this hypothesis, further studies in controlled laboratory environments are necessary to investigate whether the species identification of *S. stridulans* females varies in accuracy depending on the composition of the multicomponent male courtship signals in the presence of heterospecific signals. Considering the female preference for complex male courtship in *S. stridulans*[68], the potential trade-off between accurate species recognition/detection and preference for complexity of conspecific females would be an interesting subject of study to understand how animals evolve complex communication displays.

Our study was able to overcome numerous challenges with vibroscape analyses. In addition to successfully acquiring natural field recordings, for example, we were able to reliably and quantitatively measure acoustic characters and filter out noise. Our goal was not the quantification of the true frequency spectrum or amplitude range of our focal vibroscape. We recognize that our inexpensive vibration sensors (Piezo disks) may have variability in frequency response, especially at higher frequencies, due to the structural features (i.e. resonant peak of the metal component of Piezo disks) and the effects of spatial position between signaling animals and the sensors[92]. Moreover, despite the usefulness of filtering varying audio files, it is possible that the adaptive background noise filtering may distort some acoustic characters (e.g. spectral bandwidth, zero-crossing rate) by the variation in background noise profiles (e.g. frequency range, signal-to-noise ratio; Supplementary Table S2; Supplementary Figs. S1–S5). Thus, prior to future studies focused on frequency spectra or amplitudes of vibroscapes, inexpensive recording equipment such as that used in our study should be properly calibrated before deployment and adjustments should be made based on data collected from other types of equipment such as laser Doppler vibrometers.

Another important challenge in vibroscape analysis is the automation of sound classification. In the present study, we did not automate the classification of all detected sounds due to the presence of numerous 'unknown' substrate vibrations[14,15,18,19,93]. While the recent technical progress in artificial intelligence and machine learning algorithms for sound analysis allows for automating sound detection and classification for soundscape analysis with large audio datasets, the current techniques are largely based on the well-documented baseline data from previous manual classification[7]. Given the dearth of information about substrate-borne vibratory signals in many species, however, these techniques did not work appropriately to classify signals from vibratory recordings that contain many unknown signals. Thus, to address this challenge in the future, it is essential to put research effort into creating a well-organized 'library' of substrate vibrations of ground-dwelling animals. With research efforts towards such a database, recent progress in machine learning algorithms will accelerate the development of techniques to monitor vibroscape in diverse habitats. As progress towards this goal, we shared the unclassified vibratory signals in a Dryad repository[94] and Supplementary Audio.

In the present study, we showed that the scope of soundscape ecology can be extended to substrate-borne vibratory communication of ground-dwelling arthropods. As compared to traditional methods using collected specimens, vibroscape analysis provided more direct and detailed information about the microhabitat, seasonal/daily temporal patterns, and acoustic characteristics of vibratory signals in a local signaling environment. Through the collection of detailed information about vibratory signaling, we suggested that *Schizocosa* wolf spiders may plastically alter their signaling behaviors in response to the abundance of abiotic and/or biotic noises in the vibroscape. In particular, *S. stridulans* males may decrease the potential risk of signal interference with closely related species, *S. uetzi*, by decreasing the vibratory signal complexity.

We expect that the developed techniques in the present study will contribute to our growing knowledge of soundscape ecology. In particular, our approach will enable us to investigate the interactions among diverse biotic, abiotic, and anthropogenic vibrations in local vibroscapes by (i) the inexpensive and simple design of vibratory sensors and (ii) the general accessibility of Python codes to use or improve the automated processing of large audio dataset. Given the increasing interest in signal interference by anthropogenic noise in many species[95–102], the application of these techniques to urban soundscape ecology will provide a powerful tool to broaden our understanding of anthropogenic signal interference into substrate-borne vibratory communication of ground-dwelling arthropods; animals that make up the major taxon of many urban ecosystems[103–105]. Moreover, in future research, we expect several improvements including (i) automated classification using machine learning algorithms to address the open-set recognition problem[106] and (ii) controlled laboratory experiments to understand the behaviors of individual species. These advances will further broaden the applicability of the developed techniques and vibroscape analysis to investigate signal evolution of largely understudied communication channel,

substrate-borne vibratory signals of diverse ground-dwelling arthropods.

## Data availability

All information needed to reproduce the results of the paper is in the paper and the Supplementary Materials. Python and R codes and raw data are archived at the Dryad data repository (https://doi.org/10.5061/dryad.0gb5mkm5w). Due to the large size, original audio recording files may be requested from the authors.

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

## Acknowledgements

We would first like to thank G. Stratton for her advice and help with fieldwork. We would also like to thank K. Yarborough and M. Baker at the field station of the University of Mississippi for their support including housing at the field station. We thank M. Kim for her help with fieldwork and data analysis. The members of the animal behavior group

at UNL provided useful comments on a draft of this manuscript. The graduate committee of N. C. (D. Shizuka, K. Montooth, J. Stevens, J. DeLong, W. Wagner) provided helpful comments on this work from the experimental design to manuscript writing. Funding was provided by the National Science Foundation to E. A. H. (IOS 1556153 & IOS 1037901), the Animal Behavior Society Student Research Grant, the American Arachnological Society Student Research Grant, R. C. Lewontin Early Award from the Society for the Study of Evolution, Lewis and Clark Fund for Exploration and Field Research from the American Philosophical Society, and Emergency Summer Support for Graduate Students from the Quantitative Life Sciences Initiative at UNL to N. C.

## Author contributions

Conceptualization: N.C. & E.A.H. Methodology: N.C., P.M. & E.A.H. Investigation: N.C. & P.M. Visualization: N.C. Supervision: E.A.H. Writing—original draft: NC Writing—review & editing: N.C. & E.A.H.

## Funding

## Competing interests

The authors declare no competing interests.
