## [Peer Review File · Communications Biology]

Reviewers' comments:

Reviewer #1 (Remarks to the Author):

In the manuscript, authors describe a new approach to record and analyse forest floor vibroscape and they use the recorded data to test the hypothesis that closely related species of *Schizocosa* spiders partition their acoustic niche. More ecological approach in research on vibrational communication is crucial to understand selection pressures under realistic natural conditions. Vibroscape research from animal communication perspective is still in its infancy and the present paper is a much needed and welcoming addition. However, I feel that in the current version, the manuscript is missing some crucial information needed to allow the readers to evaluate and interpret the results. Such information is also crucial in replicating their approach elsewhere.

I suggest that the manuscript is reviewed again when the authors include the missing information.

1. My main concern relates to the post-processing and treatment of individual recordings. As stated in the Methods, there were 125 individual recording units, 25 for each of the five plots. In the text, I did not find the information, how had the authors treated the 25 individual recordings from the same plot. Did they treat 24-hour recordings from individual recording units as independent regardless of the plot identity? Does a temporal distribution of recorded vibrations shown in Fig. 3a, show vibrations recorded from a single representative plot or a temporal compilation from all five plots (figure legend does not explain this)? Did the authors find any differences between plots with the same microhabitat?

2. It is not clear whether the authors used automated procedure to analyse all recordings. Did they validate their automated approach on a subset of manually annotated recordings?

3. The number of spiders collected in the traps is fairly low (Fig. 5a); however, it is not stated whether the numbers shown are cumulative numbers for all five plots or not. If the numbers shown are cumulative, how many *Schizocosa* males of each species were collected at individual plots? For example, *S. uetzi* and *S. stridulans* may live in the same microhabitat (leaf litter, Fig. 4); however, this does not mean that they were found syntopically at all leaf litter plots.

4. As shown previously, as well as in this manuscript, airborne-sounds are also detected in the vibroscape. Because the authors recorded the vibroscape on an artificial substrate (waterproof paper) it would be helpful to expand the discussion by information, how such approach compares with the detection of airborne-sound in the recordings on the natural substrate. A large flat sheet of paper might amplify airborne sounds and make them more detectable than on a natural substrate. Is the amplitude of vibrations corresponding to bird songs comparable on the paper and on the natural substrate? Would it be high enough to be detected by spiders when recorded on the natural substrate?

5. The authors should define what they consider 'noise'. Everything that may interfere with spider vibrational communication, i.e. everything that is not *Schizocosa* vibrational signal? Do authors have any information about the frequency structure of noise? Is it possible that the observed frequency shift in dominant frequency of *Schizocosa* vibrational signals in the presence of noise is to avoid a frequency overlap, i.e. to increase detectability to conspecific females? It is not clear to me, why the authors focus on explanation of decreased detectability to eavesdroppers. Stronger attenuation of higher frequencies also results in lower detectability to intended receivers and following this argument, higher frequency would be an advantage in avoiding predation risk only if vibrational communication takes place at close range. If this is the case, the authors should explain it.

6. Taken into account that the number of females at the plots is not known, could the alternative explanation to an increased number of idles be the presence of females?

Reviewer #2 (Remarks to the Author):

Title: Vibroscape analysis reveals acoustic niche overlap and plastic alteration of vibratory courtship signals in ground-dwelling wolf spiders

The authors describe a low cost technique to record and evaluate vibrosapes across plant litter microhabitats. Using automated techniques, the authors demonstrate the technique in identifying wolf spider vibratory displays and use it to test several hypothesis regarding signaling niche use. The authors find separation in the signaling niches of some but not all wolf spider species observed. Additionally, the authors found evidence for plasticity in signaling behavior in response to environmental noise. Overall, this article is excellent. The techniques described will be of great use to bioacousticians and behavioral ecologists. Additionally, the authors show one of the ways that this technique can be used to answer important questions in behavioral ecology in ways that have not been possible before. I had a few reservations however. First, the limitations of the technique was hard to assess particularly on whether relative rarity of vibratory songs had to do with the absence of an extensive vibratory song library or the dominance of wolf spiders in the habitats measured. Justification of some of the analytical parameters used is also needed (e.g. frequency and temporal bands). Finally, I had a few concerns regarding the interpretations of the niche overlap data. I look forward to seeing this article in print! This will be a major step forward for biologists interested in animal communication.

Line 10: rewrite to "we tested the hypothesis that three closely related species of Schizocosa wolf spider..."

Line 9 -13: long sentence. Split into two

Line 13: authors only analyze dominant frequency... this sentence should reflect that

Line 14: replace "promotes" with "is related to"

Line 40: the collections and classification of vibratory sounds seems to be something necessary to improve soundscape analysis. Authors should call for more research in this area in the discussion

Line 50-51: use "Uhl, G. and Elias, D. O. (2011) Communication. In: Spider Behaviour: Versatility and Flexibility (Edited by Heberstein, M.E.), Cambridge University Press" as spider reference

Line 52: delete "and the high price of the device". This is stated elsewhere

Line 55-56: delete "biased technological progress in soundscape ecology"

Line 55-58: Technique could be useful for many purposes and this pushes unneeded constraints on the technique. Suggest broadly talking about all the uses of soundscape data including automation that would include biodiversity surveys, assessing environmental impacts, ecological interactions, species interactions, sensory ecology, impacts of noise, etc. Currently sentence is too narrow

Line 61: rewrite "sound classification algorithms" to algorithms to classify airborne sound"

Line 88-94: this is very specific to the study conducted and as such is a bit early in the introduction.

This part would be more appropriate in the last paragraph

Line 95: some review of the relevant literature regarding niche overlap in airborne acoustic signals is necessary in this introduction.

Line 127-128: unclear what the authors are referring to here. That there were known schizocosa species? That populations were dense? That there were lots of arthropods. Suggest authors be more specific as to why these areas were chosen

Line 158: The automation techniques make a lot of assumptions as to the salient features of animal signaling. Was this maximized for wolf spider recordings? Does "biologically meaningful" refer to wolf spider communication specifically?

Line 160: why ten minutes? Was this for processing speed or because wolf spider interactions occur within this time frame?

Line 167: delete "by sigma clipping"

Line 180: define bout in this context.

Line 192-193: could known spider songs be classified using the algorithm the way bird songs were?

Line 200-201: is it possible to measure the counts for different songs in the data. This technique could be extremely useful in surveying vibratory arthropods and this type of analysis would allow readers to

assess its usefulness for this context. Additionally, one could get useful information regarding the vibroscape and natural history of this habitat. There is no need to identify all species but there is a need to know how useful this technique would be if there was a vibratory sound library available.
Line 202-2014: the citations here are missing information. Some should be "et. al". This may occur at other spots in the MS

Line 223-224: why the 100Hz bin? This would seem more appropriate for airborne sounds given the relatively tonal nature of airborne sounds. Suggest more justification although I would also suggest that the data be examined with smaller bins. This is evident from looking at the plasticity in response to noise which suggests to me that the changes in frequency may be smaller and authors could be missing crucial responses

Line 237: I suggest that the analysis only include schizocosa species unless there is a compelling reason to examine the other sounds... where they other spiders? Predators? Competitors? As such there is little one can infer without more information and the comparison with schizocosa is already rich.

Line 241: define peak rate

Line 249: the use of +/- 15 minutes is awkward. Suggest explaining it explicitly and removing the constant use of the term to improve readability

Line 250-251: was there an attempt to use other signaling parameters here? Given the broadband nature of spider sounds, bandwidth, minimum frequency, and maximum frequency may be more relevant parameters. Dominant frequency is more relevant to tonal signals like those used by birds and seen in the MS sample spectrograms.

Line 253-255: what about other important factors in schizocosa signaling such as signaling rate (is this peak rate?)

Line 277: restate the noise categories here

Line 291-292: I found this confusing. Maximum occurrence within the data set? Within a day?

Line 311-321: Authors should say something about unclassified songs. How many total? What is the distribution of occurrences? In my opinion these could be one of the sources for the most novel data emerging from the ms and inform readers about its possibilities. Authors should upload unclassified sounds..

Line 343-344: how does this compare with airborne soundscapes? This should be covered in introduction and discussed in discussion

Line 346: is there any information about these other species? Order? This information while interesting can not be interpreted without more info. Suggest deleting.

Line 380-381: currently this is overstated. If authors have more information about the diversity of sounds in the vibroscape and occurrences, this would be stronger. Currently this data is rich only for schizocosa species

Line 378-380: does this mean that airborne sounds (birds) were more prevalent (excluding schizocosa)?

Line 382: replace "quantifications" with "data"

Line 385-386: rewrite to "and *S. uetzi* signaling microhabitat use (pine litter vs. leaf litter respectively)."

Line 387: delete "however"

Line 388: replace "pattern of" with "evidence for"

Line 391: replace "through variation in" with "by increasing their"

Line 391-393: rewrite to "*S. stridulans* shows plasticity in response to noise by increasing courtship signaling in terms of the number of specific components (i.e. idles) with the abundance of heterospecific *S. uetzi* signaling potentially in response to high signaling niche overlap"

Line 394: I would disagree that this study examines mechanisms of acoustic niche partitioning

Line 396-404: move paragraph to later in the discussion

Line 409: were substrate borne vibrations other than schizocosa that rare or were identifiable songs rare?

Line 405-411: other potential questions to explore: How many bouts are unknown vibratory signals? is the habitat dominated by schizocosa or unknown species? How much is substrate vs airborne signals?

Line 421-422: What about a bigger library of substrate-borne songs? Would this be more or less

important than open set recognition algorithms? Either could be very useful in terms of diversity assessments

Line 427: delete "fine" and "general"

Line 429: delete "general" and "at the moment of signal production."

Line 429-442: I was generally confused by this paragraph. Judging by the spectrograms of noise in the MS which is all low frequency, wouldn't it drive signals to be higher as found in this study. Singing at lower frequencies would increase potential signal masking. This is a pattern found in airborne signalers in response to noise. The arguments in those papers would be similar to substrate borne signalers (see Shannan et al 2016, Raboin and Elias 2019, Barber et al 2011). I am unconvinced by the predator hypothesis especially given that data on vibratory receptors suggests the ability to detect a large range of frequencies.

Line 451: delete "the"

Line 454-457: MS about schizocosa so this sentence should reflect this. Alternatively, authors can expand the discussion and include more unknown songs.

Line 458-464: Are these only evolutionary responses? What about more "plastic" responses?

Discussion would be improved by being specific about what type of response the authors are talking about

Line 490: any thoughts about closely related schizocosa species where this is not the case (i.e. *S. ocreata*)

Line 499: delete "that occurred together"

Line 501-504: rewrite to : Also, we suggested that *S. stridulans* males may decrease the potential risk of signal interference.

Line 506: replace "broadening the application of" with "studying"

Figures: many figures are superfluous and should be either deleted or moved to supplementary information: Particularly – figure 6, figure 8,

.Figure 4: remove unknown species

Figure 6: use only schiz data

Figure 9: y axis legends awkward. Fix grammar (Conspecific density Index, Heterspecific signal Index)

Table 1: remove unknown species

Table 2: fix grammar (number of noise)

Reviewer #3 (Remarks to the Author):

This is an exciting study that quantifies leaf-litter vibrosapes for the first time. The authors use an array of sensors deployed across five plots and recorded continuously for multiple days. The authors then use a custom-written automated detection software, along with comparison to known sources, to identify multiple sources of vibrational events. The study focuses on three wolf spider species, ground-truthing the acoustic results using an independent measure of spider activity obtained from pitfall traps. The results allow the authors not only to describe the spiders' signaling activity at an unprecedented spatiotemporal scale, but also to quantify the degree of acoustic niche overlap between the species. There is also an interesting finding that the spiders seem to be increasing the dominant frequency of their signals under certain noise conditions.

This study describes novel approaches and results that will be of interest to a broad readership, given the current level of interest in ecoacoustics based on airborne or waterborne sound. I have a number of minor comments and questions, and one concern that is potentially relevant to interpretation of some of the results. I hope these comments will be useful to the authors as they prepare their paper for publication.

Primary concern:

1. The finding that the spiders increase the dominant frequency of their signals under some noisy conditions is interesting and seems to parallel similar findings in other animal systems. However, I

have some concerns with the analysis that leave me unconvinced that the spiders are in fact changing the frequency content of their signals.

a. The authors need to demonstrate that their filtering algorithm preserves the original frequency information present in the signals. Because much of the noise is in lower frequencies, it is possible that filtering out the noise might also remove some of the lower-frequency energy in the signals, yielding a measurement of a higher DF – an effect that would be more pronounced at higher noise levels. A verbal rebuttal would not be sufficient to show that noise reduction is not driving the pattern of increasing dominant frequency in spider signals. However, I would be satisfied if the authors could take clean signals, add varying amounts of noise, apply their filters, and show before and after spectra and/or measurements.

b. In both panels in figure 7b, the extreme low or high values are on one side of the graph where there are few signals. Without more information, it is hard to rule out the possibility that these represent a different subset of spiders using e.g., marginal habitat or different substrate conditions.

c. Furthermore, for 7b left panel, the significant effects are very low in magnitude (near zero slope), raising the question of whether, even if the measurements are accurate, the differences are biologically meaningful.

Other issues

2. In the key figures, the data from all the recorders are pooled. This is appropriate for showing the overall patterns, but these figures do not reveal the signaling environments that individual spiders encounter. Perhaps the authors could add a figure illustrating short-term results, even from single sensors? When recordings made over a large area are pooled it looks like there is a lot of potential for signal interference, but how often are spiders actually signaling within the auditory range of another signaling spider?

3. More details of the recording setup are needed. For example, a search for the recorder used (Toobom R01 8GB acoustic recorder) yields a model with built-in microphones rather than an input for an external microphone. Did the authors use a model with a mic input? If not, were the units modified? Did the input have a suitably high impedance input for a piezo sensor or was some other input stage necessary?

4. The authors should mention the pros and cons of inexpensive contact microphones. Similar piezo mics have been used in previous studies and they are suitable for recording the presence of signals, their temporal features and the overall frequency range. However, they are not calibrated sensors, and some care must be taken when interpreting the frequency/amplitude characteristics of those signals. These issues of tradeoffs between expensive and inexpensive vibration sensors are discussed in Nieri et al (2022

Inexpensive Methods for Detecting and Reproducing Substrate-Borne Vibrations: Advantages and Limitations).

5. The absorption of airborne sound by leaves and leaf litter is a very well-known phenomenon and its relevance to vibrational communication has been pointed out in previous studies. When discussing this result the authors should cite (at least) the Sturm et al (2021) paper, which they cite elsewhere and which abundantly documents the presence of substrate vibrations sourced in airborne sound, and a study on leaf-litter wolf spiders that highlights the impact of airborne sound on their behavior (<https://doi.org/10.1093/beheco/ars016>)

6. The finding that *S. stridulans* has more idles in the presence of conspecifics & heterospecifics is interesting, and the focus on idles is relevant to spider social behavior. In Figure 9, it would be helpful to show the data points in addition to the model predictions.

7. Some of the axis labels in figures use non-standard terms. For example, the x-axis labels for two panels of Figure 7 are “number of noise.” The X axis labels should be edited to reflect the terms used in text.

Overall, this study involves a substantial amount of work with an efficient computational workflow to analyze large vibroscape datasets. Prior to publication, we recommend more critically assessing how the moving noise baseline and filtering impacts the data, particularly when extracting and modeling spectral characteristics within spider signals.

Author Response: Thank you for the opportunity to revise our manuscript. In the responses below, our responses to reviewer comments can be found in blue.

Important updates:

1. For the analysis of the effects of general noise and conspecific/heterospecific signals on signal properties, we only used the dominant frequency due to the high distortion of other signal properties by our adaptive background noise filtering.
2. For the analysis of the variation in signal complexity of *S. stridulans* (Figure 7), we changed the dependent variable from the number of idles to the duration of idles because the number of idles may not be an accurate measure due to the potential impacts of the pulse grouping algorithms on the number of idles.

Figure updated:

1. We removed Figure 4 in the previous version due to the limitations of visualizations (up to 10 figures and tables). Instead, we added the information in Table 1a.
2. Figure 5 and Figure 6 about the plastic alteration of signal properties by general noise and conspecific/heterospecific signals have been changed due to the change in the results.
3. Figure 7 about the effects of the abundance of heterospecific signals on the vibratory signal complexity of *S. stridulans* has been changed due to the change in the analysis (the number of idles to the duration of idles).

Reviewers' comments:

Reviewer #1:

In the manuscript, authors describe a new approach to record and analyse forest floor vibroscape and they use the recorded data to test the hypothesis that closely related species of Schizocosa spiders partition their acoustic niche. More ecological approach in research on vibrational communication is crucial to understand selection pressures under realistic natural conditions. Vibroscape research from animal communication perspective is still in its infancy and the present paper is a much needed and welcoming addition. However, I feel that in the current version, the manuscript is missing some crucial information needed to allow the readers to evaluate and interpret the results. Such information is also crucial in replicating their approach elsewhere.

I suggest that the manuscript is reviewed again when the authors include the missing information.

1. My main concern relates to the post-processing and treatment of individual recordings. As stated in the Methods, there were 125 individual recording units, 25 for each of the five plots. In the text, I did not find the information, how had the authors treated the 25 individual recordings from the same plot. Did they treat 24-hour recordings from individual recording units as independent regardless of the plot identity? Does a temporal distribution of recorded vibrations shown in Fig. 3a, show vibrations recorded from a

single representative plot or a temporal compilation from all five plots (figure legend does not explain this)? Did the authors find any differences between plots with the same microhabitat?

- We treated the recordings from individual recording units as independent events to each other because the distance between the recording units is beyond the distance that substrate-borne vibratory signals of *Schizocosa* wolf spiders are known to travel (2 meters; Uetz et al., 2013).
- Across the plots, the signals of *Schizocosa stridulans* and *S. uetzi* occurs differently between plot A and other leaf litter plots (Supplementary Table S1). However, for the analysis of spatial overlap, we analyzed the spatial niche overlap based on the plots, not microhabitat types, so the variation across plots did not affect the further results. We clarified the description of spatial niche overlap in the method section (Line 218). Thank you for pointing this out.
- In Figure 3a, we included all the detections from every recording plot for visualization of temporal patterns. We added supplementary figures showing temporal patterns in each recording plot for further information (Supplementary Figure S6-S10).

2. It is not clear whether the authors used automated procedure to analyse all recordings. Did they validate their automated approach on a subset of manually annotated recordings?

- Yes. We reported the verification methods and results in the Supplementary Materials S1.

3. The number of spiders collected in the traps is fairly low (Fig. 5a); however, it is not stated whether the numbers shown are cumulative numbers for all five plots or not. If the numbers shown are cumulative, how many *Schizocosa* males of each species were collected at individual plots? For example, *S. uetzi* and *S. stridulans* may live in the same microhabitat (leaf litter, Fig. 4); however, this does not mean that they were found syntopically at all leaf litter plots.

- The numbers in Table 1a (Figure 4 was deleted due to the limited number of visualizations) are the cumulative counts of *Schizocosa* males across plots. Due to the small number of collected specimens, we cannot do a proper statistical analysis to test the variation in data from the pitfall trapping across plots, but *S. stridulans* and *S. uetzi* were collected together from all the leaf litter plots. We added a table in the supplementary material (Supplementary Table S1).

4. As shown previously, as well as in this manuscript, airborne-sounds are also detected in the vibroscope. Because the authors recorded the vibroscope on an artificial substrate (waterproof paper) it would be helpful to expand the discussion by information, how such approach compares with the detection of airborne-sound in the recordings on the natural substrate. A large flat sheet of paper might amplify airborne sounds and make them more detectable than on a natural substrate. Is the amplitude of vibrations corresponding to bird songs comparable on the paper and on the natural substrate? Would it be high enough to be detected by spiders when recorded on the natural substrate?

- We did not have a dataset to compare the amplitude of airborne sound between artificial and natural substrates in our study. A previous study (Gordon & Uetz, 2012) showed that seismic interference by bird songs in a leaf litter habitat reduces male courtship behaviors and mating success of another *Schizocosa* wolf spider species (*S. ocreata*). This result suggests that the amplitude of substrate-borne vibration by airborne sounds on our study plots may be sufficient to influence the behavior of our study species. We discussed this point in the discussion (Line 396-400).
- Line 396 – 400: “While the transmission to the artificial substrate (i.e. waterproof paper) in our recording device may be different in acoustic characters (e.g. amplitude) as compared to what animals experience in natural vibroscope, a previous study suggested that airborne sounds

transmitted to the ground influence the substrate-borne communication of another ground-dwelling wolf spider (*Schizocosa ocreata* – Gordon & Uetz, 2012).”

5. The authors should define what they consider ‘noise’. Everything that may interfere with spider vibrational communication, i.e. everything that is not *Schizocosa* vibrational signal? Do authors have any information about the frequency structure of noise? Is it possible that the observed frequency shift in dominant frequency of *Schizocosa* vibrational signals in the presence of noise is to avoid a frequency overlap, i.e. to increase detectability to conspecific females? It is not clear to me, why the authors focus on explanation of decreased detectability to eavesdroppers. Stronger attenuation of higher frequencies also results in lower detectability to intended receivers and following this argument, higher frequency would be an advantage in avoiding predation risk only if vibrational communication takes place at close range. If this is the case, the authors should explain it.

- The authors should define what they consider ‘noise’. Everything that may interfere with spider vibrational communication, i.e. everything that is not *Schizocosa* vibrational signal?

Yes. We added the definition of noise for our analysis for Obj. IIIa (Line 277-278).

- Line 277-278: “... general noise (airborne + substrate-borne abiotic/biotic noise) that occurred during the interaction time window of each signal bout ...”

- Do authors have any information about the frequency structure of noise? Is it possible that the observed frequency shift in dominant frequency of *Schizocosa* vibrational signals in the presence of noise is to avoid a frequency overlap, i.e. to increase detectability to conspecific females?

In the revised analysis, we could not find a significant change in dominant frequency by abiotic/biotic noise.

6. Taken into account that the number of females at the plots is not known, could the alternative explanation to an increased number of idles be the presence of females?

- We changed the analysis to compare the actual duration of idles. In the revised analysis, males reduced the duration of idles responding to the abundance of heterospecific signals. We provided explanations about the revised results in the discussion (Line 471-482).

Reviewer #2:

The authors describe a low cost technique to record and evaluate vibrosapes across plant litter microhabitats. Using automated techniques, the authors demonstrate the technique in identifying wolf spider vibratory displays and use it to test several hypothesis regarding signaling niche use. The authors find separation in the signaling niches of some but not all wolf spider species observed. Additionally, the authors found evidence for plasticity in signaling behavior in response to environmental noise. Overall, this article is excellent. The techniques described will be of great use to bioacousticians and behavioral ecologists. Additionally, the authors show one of the ways that this technique can be used to answer important questions in behavioral ecology in ways that have not been possible before. I had a few reservations however. First, the limitations of the technique was hard to assess particularly on whether relative rarity of vibratory songs had to do with the absence of an extensive vibratory song library or the dominance of wolf spiders in the habitats measured. Justification of some of the analytical parameters used is also needed (e.g. frequency and temporal bands). Finally, I had a few concerns regarding the

interpretations of the niche overlap data. I look forward to seeing this article in print! This will be a major step forward for biologists interested in animal communication.

1. Line 10: rewrite to “we tested the hypothesis that three closely related species of *Schizocosa wolf* spider...”

- We rewrote the sentence (Line 6–9).
- Line 6-9: “We tested (i) the acoustic niche partitioning and (ii) plastic behavioral response to reduce the risk of signal interference in the presence of substrate-borne noise (e.g. anthropogenic noise) and conspecific/heterospecific signals in ground-dwelling *Schizocosa wolf* spiders.”

2. Line 9 -13: long sentence. Split into two

- We removed the sentence.

3. Line 13: authors only analyze dominant frequency... this sentence should reflect that

- We changed the sentence (Line 11).

4. Line 14: replace “promotes” with “is related to”

- We deleted the sentence.

5. Line 40: the collections and classification of vibratory sounds seems to be something necessary to improve soundscape analysis. Authors should call for more research in this area in the discussion.

- We added sentences about the importance of research efforts on the ‘library’ of substrate-borne vibrations from animals in the discussion (Line 509-514).
- Line 509-514: “Thus, to address this challenge in the future, it is essential to put research effort into creating a well-organized ‘library’ of vibratory sounds of ground-dwelling animals. With research efforts towards such a database, recent progress in machine learning algorithms will accelerate the development of techniques to monitor vibroscape in diverse habitats. As progress towards this goal, we shared the unclassified vibratory signals in a Dryad repository (Choi et al., 2023).”

6. Line 50-51: use “Uhl, G. and Elias, D. O. (2011) Communication. In: *Spider Behaviour: Versatility and Flexibility* (Edited by Heberstein, M.E.), Cambridge University Press” as spider reference

- We added the reference. Thank you for your suggestions (Line 46–47).

7. Line 52: delete “and the high price of the device”. This is stated elsewhere

- Deleted (Line 48)

8. Line 55-56: delete “biased technological progress in soundscape ecology”

- Deleted (Line 51)

9. Line 55-58: Technique could be useful for many purposes and this pushes unneeded constraints on the technique. Suggest broadly talking about all the uses of soundscape data including automation that would include biodiversity surveys, assessing environmental impacts, ecological interactions, species interactions, sensory ecology, impacts of noise, etc. Currently sentence is too narrow.

- We revised the sentence (Line 51-57)
- Line 51-57: “This arguably impedes our broader understanding of how ecological and environmental interactions with biological, geophysical, and anthropogenic vibrations in natural habitats influence vibratory communications and their functions. Such an understanding is critical to our understanding of animal communication, however, as these interactions could promote evolutionary changes in the spatial, temporal, and acoustical properties of communications of many species involved in the vibrosapes.”

10. Line 61: rewrite “sound classification algorithms” to “algorithms to classify airborne sound”

- Changed (Line 59)

11. Line 88-94: this is very specific to the study conducted and as such is a bit early in the introduction. This part would be more appropriate in the last paragraph

- We revised the sentence (Line 85-94)
- Line 85-94: “In particular, as compared to the traditional survey by manual observation, soundscape analysis using autonomous sensors can enable researchers to use detailed information about variations in local soundscapes in various spatial and temporal scales to investigate how animals respond to short-term or long-term exposure to abiotic and biotic noise and acoustic niche overlap (Shonfield & Bayne, 2017; van Opzeeland & Boebel 2018; Gottesman et al., 2018; Kleyn et al., 2021; Hart et al., 2021; Ross et al., 2023). These data, in turn, can be used to directly test our understanding of ecological and evolutionary processes, such as the effects of abiotic or biotic noise disrupting the acoustic niche partitioning on signal evolution by reproductive character displacement (Pfennig & Pfennig, 2009) and/or behavioral plasticity (Halfwerk et al., 2019; Wong & Candolin, 2015).”

12. Line 95: some review of the relevant literature regarding niche overlap in airborne acoustic signals is necessary in this introduction.

- We added more references related with the use of passive monitoring for understanding acoustic niche hypothesis (Line 89-90)

13. Line 127-128: unclear what the authors are referring to here. That there were known Schizocosa species? That populations were dense? That there were lots of arthropods. Suggest authors be more specific as to why these areas were chosen

- We revised the sentence (Line 127-128)
- Line 127-128: “Before we chose study plot locations, we checked for the presence of matured and immature ground-dwelling wolf spiders by direct observation (Figure 1a).”

14. Line 158: The automation techniques make a lot of assumptions as to the salient features of animal signaling. Was this maximized for wolf spider recordings? Does “biologically meaningful” refer to wolf spider communication specifically?

- Our programs optimally altered the threshold for automated detection based on the values (e.g. amplitude, intervals between detected pulses) of each file. However, it is not certain that the optimization of features is also “biologically meaningful”, so we deleted the expression in the sentence (Line 162-163).

15. Line 160: why ten minutes? Was this for processing speed or because wolf spider interactions occur within this time frame?

- There is no specific reason besides the processing speed. Also, the size of the chunks does not have any effect on further analysis. We revised the sentence to avoid confusion (Line 163-164).
- Line 163-164: “Before the process, we divided each 24-hour recording WAV file into 10-minute chunks using FFmpeg (Tomar, 2006) for processing speed.”

16. Line 167: delete “by sigma clipping”

- Deleted (Line 170)

17. Line 180: define bout in this context.

- We added our definition of the bout (Line 188-189).

18. Line 192-193: could known spider songs be classified using the algorithm the way bird songs were?

- It is possible to apply the algorithm for the identification of spider signals. However, significant research efforts should proceed to acquire datasets and feature selection for training models which is beyond our research scope in this manuscript.

19. Line 200-201: is it possible to measure the counts for different songs in the data? This technique could be extremely useful in surveying vibratory arthropods and this type of analysis would allow readers to assess its usefulness for this context. Additionally, one could get useful information regarding the vibroscape and natural history of this habitat. There is no need to identify all species but there is a need to know how useful this technique would be if there was a vibratory sound library available.

- Yes. We counted the number of different types of ‘unknown’ sounds in our dataset. We will provide the spectrograms and sample audios in the Dryad repository (Choi et al., 2023 - Supplementary Material S3). We also discussed the importance of a vibratory sound library for further studies (Line 509-513).
- Line 509-514: “Thus, to address this challenge in the future, it is essential to put research effort into creating a well-organized ‘library’ of vibratory sounds of ground-dwelling animals. With research efforts towards such a database, recent progress in machine learning algorithms will accelerate the development of techniques to monitor vibroscape in diverse habitats. As progress towards this goal, we shared the unclassified vibratory signals in a Dryad repository (Choi et al., 2023).”

20. Line 202-204: the citations here are missing information. Some should be “et. al”. This may occur at other spots in the MS

- We fixed the error. Thank you for pointing it out.

21. Line 223-224: why the 100Hz bin? This would seem more appropriate for airborne sounds given the relatively tonal nature of airborne sounds. Suggest more justification although I would also suggest that the data be examined with smaller bins. This is evident from looking at the plasticity in response to noise which suggests to me that the changes in frequency may be smaller and authors could be missing crucial responses

- We decreased the frequency bin size to 10 Hz.

22. Line 237: I suggest that the analysis only include Schizcosa species unless there is a compelling reason to examine the other sounds... where they other spiders? Predators? Competitors? As such there is little one can infer without more information and the comparison with Schizocosa is already rich.

- We excluded the unknown sounds from the analysis. Thank you for your suggestion.

23. Line 241: define peak rate

- We removed the analysis of the peak rate in the revised analysis.

24. Line 249: the use of +- 15 minutes is awkward. Suggest explaining it explicitly and removing the constant use of the term to improve readability

- We changed the term to the *interaction time window* (Line 244).

25. Line 250-251: was there an attempt to use other signaling parameters here? Given the broadband nature of spider sounds, bandwidth, minimum frequency, and maximum frequency may be more relevant parameters. Dominant frequency is more relevant to tonal signals like those used by birds and seen in the MS sample spectrograms.

- We tested the potential use of other parameters (spectral centroids, spectral bandwidth, zero-crossing rates), but, unfortunately, those characteristics were significantly influenced by the variation in background noise profiles that were used for our filtering method (Supplementary Materials S1). We discussed this limitation of our study (Line 260-263; Line 489-500).
- Line 260-263: “We chose these two characters due to their relative robustness to our noise filtering method which may vary other acoustic characters depending on the background noise profiles (e.g. frequency range, signal-to-noise ratio; Supplementary Materials S1)”
- Line 489-500: “Our goal was not the quantification of the true frequency spectrum or amplitude range of our focal vibroscape. We recognize that our inexpensive vibration sensors (Piezo disks) may have variability in frequency response, especially at higher frequencies, due to the structural features (i.e. resonant peak of the metal component of Piezo disks) and the effects of spatial position between signaling animals and the sensors (Nieri et al., 2022). Moreover, despite the usefulness of filtering varying audio files, it is possible that the adaptive background noise filtering may distort some acoustic characters (e.g. spectral bandwidth, zero-crossing rate) by the variation in background noise profiles (e.g. frequency range, signal-to-noise ratio; Supplementary Material S1). Thus, prior to future studies focused on frequency spectra or amplitudes of vibroscares, inexpensive recording equipment such as that used in our study should be properly calibrated before deployment and adjustments made based on data collected from other types of equipment such as laser Doppler vibrometers.”

26. Line 253-255: what about other important factors in Schizocosa signaling such as signaling rate (is this peak rate?)

- Thank you for your suggestion. However, to quantify the signaling rate, it is necessary to know the identity of signalers of all detected signals. Unfortunately, it is currently not possible to acquire the data in our dataset.

27. Line 277: restate the noise categories here

- We added the definition of general noise (Line 277-278).

28. Line 291-292: I found this confusing. Maximum occurrence within the data set? Within a day?

- We used the maximum occurrence within the dataset. We clarified it in the sentence (Line 284-287).

- Line 284-287: “To quantify the relative abundance, we divided the number of conspecific/heterospecific signals during the interaction time window of each signal bout by the maximum value across the whole recording period (*S. stridulans*: 7 times, *S. uetzi*: 11 times) so that the value ranges from 0 to 1.”

29. Line 311-321: Authors should say something about unclassified songs. How many total? What is the distribution of occurrences? In my opinion these could be one of the sources for the most novel data emerging from the ms and inform readers about its possibilities. Authors should upload unclassified sounds.

- We introduced the current limitation in sound classification and share the unclassified signals in the Dryad repository (Supplementary Materials S3) (Line 315-316).

30. Line 343-344: how does this compare with airborne soundscapes? This should be covered in introduction and discussed in discussion.

- We compared the Pianka index between *S. stridulans* and *S. uetzi* with that of previous research on anurans and avians (Line 410-415).

31. Line 346: is there any information about these other species? Order? This information while interesting can not be interpreted without more info. Suggest deleting.

- Unfortunately, we currently do not have any information about the unknown vibration. We deleted the sentence.

32. Line 380-381: currently this is overstated. If authors have more information about the diversity of sounds in the vibroscape and occurrences, this would be stronger. Currently this data is rich only for *Schizocosa* species

- Agreed. We revised the sentence (Line 384) and added a discussion about the necessity of further investigation of the diversity of vibrosapes (Line 509-514).

33. Line 378-380: does this mean that airborne sounds (birds) were more prevalent (excluding *Schizocosa*)?

- In our dataset, yes. However, this does not mean the airborne soundscape is more diverse because the airborne sounds can be collected by multiple sensors in contrast to vibratory signals that can be collected at in very close distance from our vibratory sensors.

34. Line 382: replace “quantifications” with “data”

- Changed (Line 385)

35. Line 385-386: rewrite to “and *S. uetzi* signaling microhabitat use (pine litter vs. leaf litter respectively).”

- Changed (Line 405-407)

36. Line 387: delete “however”

- Deleted (Line 407)

37. Line 388: replace “pattern of” with “evidence for”

- We removed the sentence in the revised version.

38. Line 391: replace “through variation in” with “by increasing their”

- We removed the sentence in the revised version.

39. Line 391-393: rewrite to “*S. stridulans* shows plasticity in response to noise by increasing courtship signaling in terms of the number of specific components (i.e. idles) with the abundance of heterospecific *S. uetzi* signaling potentially in response to high signaling niche overlap”

- Changed (Line 470-471)

40. Line 394: I would disagree that this study examines mechanisms of acoustic niche partitioning.

- Agreed. We toned down the sentence (Line 385-387).
- Line 385-387: “... we were able to test hypotheses about acoustic niche overlap among closely related species and the behavioral responses of multimodal signaling wolf spiders (*Schizocosa duplex*, *S. stridulans*, and *S. uetzi*).”

41. Line 396-404: move paragraph to later in the discussion

- Moved (Line 390-394)

42. Line 409: were substrate borne vibrations other than *Schizocosa* that rare or were identifiable songs rare?

- In our dataset, ~ 40 % of detected substrate-borne vibrations were from *Schizocosa* wolf spiders (1627/3935). We detected 1817 signal bouts of one of the ‘unknown’ substrate-borne vibrations.

43. Line 405-411: other potential questions to explore: How many bouts are unknown vibratory signals? is the habitat dominated by *Schizocosa* or unknown species? How much is substrate vs airborne signals?

- We focused on the analysis of signals of the *Schizocosa* wolf spider because the comparison with airborne sounds cannot be addressed properly due to the variation in detectability between substrate-borne and airborne sounds. We discussed the necessity of further analysis of unknown signals in discussion (Line 509-514).

44. Line 421-422: What about a bigger library of substrate-borne songs? Would this be more or less important than open-set recognition algorithms? Either could be very useful in terms of diversity assessments

- We suggested the necessity of a sound library in the discussion (Line 509-514). Thank you for the suggestions.

45. Line 427: delete “fine” and “general”

- Deleted (Line 454)

46. Line 429: delete “general” and “at the moment of signal production.”

- Deleted (Line 455)

47. Line 429-442: I was generally confused by this paragraph. Judging by the spectrograms of noise in the MS which is all low frequency, wouldn't it drive signals to be higher as found in this study? Singing at lower frequencies would increase potential signal masking. This is a pattern found in airborne signalers in

response to noise. The arguments in those papers would be similar to substrate-borne signalers (see Shannan et al 2016, Raboin and Elias 2019, Barber et al 2011). I am unconvinced by the predator hypothesis especially given that data on vibratory receptors suggests the ability to detect a large range of frequencies.

- We removed the paragraph because we did not find a significant change in dominant frequency in the revised analysis.

48. Line 451: delete “the”

- Deleted (Line 422)

49. Line 454-457: MS about Schizocosa so this sentence should reflect this. Alternatively, authors can expand the discussion and include more unknown songs.

- We deleted the sentence.

50. Line 458-464: Are these only evolutionary responses? What about more “plastic” responses? Discussion would be improved by being specific about what type of response the authors are talking about

- We revised the paragraph (Line 428-449).

51. Line 490: any thoughts about closely related Schizocosa species where this is not the case (i.e. *S. ocreata*)

- We deleted the sentence.

52. Line 499: delete “that occurred together”

- Deleted (Line 520)

53. Line 501-504: rewrite to: Also, we suggested that *S. stridulans* males may decrease the potential risk of signal interference.

- Changed (Line 521-525)
- Line 521-525: “...we suggested that Schizocosa wolf spiders may plastically alter their signaling behaviors in response to the abundance of abiotic and/or biotic noises in the vibroscape. In particular, *S. stridulans* males may decrease the potential risk of signal interference with closely related species, *S. uetzi*, by decreasing the vibratory signal complexity.”

54. Line 506: replace “broadening the application of” with “studying”

- Changed (Line 526)

55. Figures: many figures are superfluous and should be either deleted or moved to supplementary information: Particularly – figure 6, figure 8,

- We deleted Figure 6 and revised Figure 8 (now, figure 6).

56. Figure 4: remove unknown species

- Removed.

57. Figure 6: use only Schiz data

- We removed Figure 6 because it is not useful to show correlations between three data points.

58. Figure 9: y axis legends awkward. Fix grammar (Conspecific density Index, Heterspecific signal Index)

- We changed the figure (Figure 7).

59. Table 1: remove unknown species

- We removed the unknown species.

60. Table 2: fix grammar (number of noise)

- Changed (Table 2)

Reviewer #3:

This is an exciting study that quantifies leaf-litter vibrosapes for the first time. The authors use an array of sensors deployed across five plots and recorded continuously for multiple days. The authors then use a custom-written automated detection software, along with comparison to known sources, to identify multiple sources of vibrational events. The study focuses on three wolf spider species, ground-truthing the acoustic results using an independent measure of spider activity obtained from pitfall traps. The results allow the authors not only to describe the spiders' signaling activity at an unprecedented spatiotemporal scale, but also to quantify the degree of acoustic niche overlap between the species. There is also an interesting finding that the spiders seem to be increasing the dominant frequency of their signals under certain noise conditions.

This study describes novel approaches and results that will be of interest to a broad readership, given the current level of interest in ecoacoustics based on airborne or waterborne sound. I have a number of minor comments and questions, and one concern that is potentially relevant to interpretation of some of the results. I hope these comments will be useful to the authors as they prepare their paper for publication.

Primary concern:

1. The finding that the spiders increase the dominant frequency of their signals under some noisy conditions is interesting and seems to parallel similar findings in other animal systems. However, I have some concerns with the analysis that leave me unconvinced that the spiders are in fact changing the frequency content of their signals.

a. The authors need to demonstrate that their filtering algorithm preserves the original frequency information present in the signals. Because much of the noise is in lower frequencies, it is possible that filtering out the noise might also remove some of the lower-frequency energy in the signals, yielding a measurement of a higher DF – an effect that would be more pronounced at higher noise levels. A verbal rebuttal would not be sufficient to show that noise reduction is not driving the pattern of increasing dominant frequency in spider signals. However, I would be satisfied if the authors could take clean signals, add varying amounts of noise, apply their filters, and show before and after spectra and/or measurements.

- We tested the effects of our filtering algorithm on the acoustic characteristics and demonstrated the data in Supplementary Materials S1 (Line 165-166). Thank you for your suggestions.

b. In both panels in figure 7b, the extreme low or high values are on one side of the graph where there are few signals. Without more information, it is hard to rule out the possibility that these represent a different subset of spiders using e.g., marginal habitat or different substrate conditions.

- We measured the acoustic characters again and found no significant effects in Figure 7. We removed the original Figure 7.

c. Furthermore, for 7b left panel, the significant effects are very low in magnitude (near zero slope), raising the question of whether, even if the measurements are accurate, the differences are biologically meaningful.

- We revised the measurement of the acoustic characteristics and did not find any significant changes in dominant frequency. We presented how to measure the acoustic characteristics through Jupyter Notebook in Supplementary Materials.

Other issues

2. In the key figures, the data from all the recorders are pooled. This is appropriate for showing the overall patterns, but these figures do not reveal the signaling environments that individual spiders encounter. Perhaps the authors could add a figure illustrating short-term results, even from single sensors? When recordings made over a large area are pooled it looks like there is a lot of potential for signal interference, but how often are spiders actually signaling within the auditory range of another signaling spider?

- We added information about the occurrence of general noise and conspecific/heterospecific signals (average, maximum, minimum occurrence) at each recording unit before and after 15 minutes of detected signal bouts of *Schizocosa* wolf spiders (Line 343-348).
- Line 343-348: “The vibroscape data showed that *Schizocosa* wolf spiders encounter abiotic/biotic noise during their signaling that may potentially induce signal interference. At each recording unit, on average, 10.624 general noises (abiotic + biotic noise; maximum = 37, minimum = 1), 3.274 conspecific signals (maximum = 12, minimum = 0), and 1.212 heterospecific signals (maximum = 11, minimum = 0) occurred before and after 15 minutes of *Schizocosa* wolf spider’s signal bout.”

3. More details of the recording setup are needed. For example, a search for the recorder used (Toobom R01 8GB acoustic recorder) yields a model with built-in microphones rather than an input for an external microphone. Did the authors use a model with a mic input? If not, were the units modified? Did the input have a suitably high impedance input for a piezo sensor or was some other input stage necessary?

- We used the recorder with an external input from contact microphone. However, unfortunately, we could not find detailed data such as impedance about the model because the manufacturer did not provide such information in the manual or the functional website.

4. The authors should mention the pros and cons of inexpensive contact microphones. Similar piezo mics have been used in previous studies and they are suitable for recording the presence of signals, their temporal features and the overall frequency range. However, they are not calibrated sensors, and some care must be taken when interpreting the frequency/amplitude characteristics of those signals. These issues of tradeoffs between expensive and inexpensive vibration sensors are discussed in Nieri et al (2022)

Inexpensive Methods for Detecting and Reproducing Substrate-Borne Vibrations: Advantages and Limitations).

- We mentioned the pros and cons of our recording setup in the discussion (Line 489-500). Thank you for your suggestions.

5. The absorption of airborne sound by leaves and leaf litter is a very well-known phenomenon and its relevance to vibrational communication has been pointed out in previous studies. When discussing this result the authors should cite (at least) the Sturm et al (2021) paper, which they cite elsewhere and which abundantly documents the presence of substrate vibrations sourced in airborne sound, and a study on leaf-litter wolf spiders that highlights the impact of airborne sound on their behavior (<https://doi.org/10.1093/beheco/ars016>)

- We added the citations. Thank you for your suggestions (Line 394-396).

6. The finding that *S. stridulans* has more idles in the presence of conspecifics & heterospecifics is interesting, and the focus on idles is relevant to spider social behavior. In Figure 9, it would be helpful to show the data points in addition to the model predictions.

- We added data points in Figure 9 (now Figure 7).

7. Some of the axis labels in figures use non-standard terms. For example, the x-axis labels for two panels of Figure 7 are “number of noise.” The X axis labels should be edited to reflect the terms used in text.

- We edited the figures.

Overall, this study involves a substantial amount of work with an efficient computational workflow to analyze large vibroscape datasets. Prior to publication, we recommend more critically assessing how the moving noise baseline and filtering impacts the data, particularly when extracting and modeling spectral characteristics within spider signals.

- We added the results of verification in Supplementary Material S1.

Reviewers' comments:

Reviewer #1 (Remarks to the Author):

The revised version of the manuscript is significantly improved. However, there are still some points mentioned below that require further clarification.

1. The authors have added a definition of noise, as suggested. However, this definition (airborne+substrate-borne, abiotic/biotic noise) only refers to what they call "general noise". Four different noise terms are used in the paper: Background noise (line 166), general noise (lines 235, 276), non-biological background noise and local background noise. The latter two terms are used in the Supplement ppt, where non-biological background noise is defined as "noise originating from recording equipment". Since "noise" or "noisiness" play such an important role in this paper, I believe that more detailed information on each term needs to be provided in the text, including the definition of abiotic and biotic noise. For example, two examples of recorded airborne sounds are shown in Fig. 2, namely 'airplane' and "unknown noise". The former is referred to in the text as "anthropogenic noise". Should this type of noise also be included in the definition of general noise? If the authors define noise as anything that can interfere with the vibrational communication of spiders, then heterospecific signals are part of general noise. Are the heterospecific signals (Objective III) only *Schizocosa* vibrational signals, all heterospecific vibrational signals, or all communication signals recorded in the substrate, including bird songs (see lines 346-348). How did the authors count the noise? It would be helpful if they could define abundance and diversity (lines 241-242). In the results, it would be helpful if the authors mentioned which were the most common sources of abiotic and biotic noise in their recordings.

2. With the advent of biotremology as a distinct discipline separate from bioacoustics (Hill & Wessel 2016; Hill et al. 2019), the terminology as "vibratory sound" is misleading. It should be either airborne sound or substrate vibrations. Vibrations induced by airborne sound sources in the substrate are still substrate vibrations. Vibroscape is a vibrational, not an acoustic environment (lines 389-390). Check the entire text for consistency.

3. Line 47: Insert reference Sturm et al. 2021

4. Line 49: Insert reference Sturm et al. 2019

5. Some references used in the text are missing in the reference list (e.g. Gross et al. 2010, Schwartz & Bee 2013, Sturm et al. 2019). Check the entire text.

6. Lines 142-143: change to... "would be simultaneously picking up a single individual".

7. Lines 160-163: if I understand correctly, all 17 "sounds" shown in Fig. 2 were detected, but not necessarily identified by automatic signal detection as well. Were the spider signals classified by visual inspection of the spectrograms (lines 194-195)? Bird songs were identified by BirdNET (lines 195-198). How were other species shown in Fig. 2 classified?

8. Line 325: delete one 'the'.

9. Line 341: 'Duplex' in italics.

10. Line 376: "Unprecedented" is a very strong word. Could it be replaced by another one?

11. Lines 397-403: I suggest rewording the text and also including the reference Lohrey et al. 2009. It should also be mentioned in the discussion that these two studies (Gordin & Uetz 2012, Lohrey et al.

2009) were conducted in the laboratory on artificial substrates (plastic/paper).

12. Is there any reference to support the hypothesis that arthropods can adjust the spectral properties of their vibrational signals in the short term?

13. Line 470: Insert space between *S. stridulans*.

14. Lines 471-474: What is meant by "...heterospecific signals at the moment of signal production"?

15. Lines 502-515: The importance of a reference library of vibrational signals has been recognized previously, and authors should cite some of the earlier work discussing this issue (e.g. Sturm et al. 2019, 2021, 2022; Frommolt et al. 2019; Akassou et al. 2022).

Reviewer #2 (Remarks to the Author):

Title: Vibroscape analysis reveals acoustic niche overlap and plastic alteration of vibratory courtship signals in ground-dwelling wolf spiders

The authors have improved the manuscript and adequately dealt with my concerns. A few, very minor thoughts and points are detailed below. I look forward to seeing this in print!

Line 19: rewrite to "...to acquire in many groups."

Line 24-26: suggest deleting this sentence. I do not think that the challenges stated in the previous sentence are any worse in modalities that humans cannot detect. Do the authors mean to say that these problems are exacerbated because they are multiplied by detection difficulties?

Line 47-50: limited detection range will also be a problem with individual sensors. Suggest rewriting to "limited detection range of a single device...". Suggest emphasizing that the large scale is possible through the use of multiple devices later.

Line 76: suggest mentioning temporal properties as manuscript includes discussion of this

Line 86: delete "can"

Line 102: sympatric, closely

Line 106: "variation in signaling behaviors"

Line 126: mature

Line 142: rewrite to "to travel (Uetz et al., 2013) thus reducing the possibility that..."

Line 177: is plotting the changes in noise level informative? It may be interesting to show changes in overall "noise" or amplitude levels across a day particularly as things such as wind noise could be an important constraint on signaling behavior

Line 259-262: I am confused by this sentence. Would noise filtering distort these other measurements?

Line 312: what is the cricket species? List if known

Line 341: duplex not italicized

Line 349-368: does this suggest different forms of plasticity are at play in regards to hetero and conspecific interference? This is very interesting to think about.

Line 378: But aren't these known because they have been identified as Schizocosa species? Suggest rewriting

Line 415: Are these species diverging in morphological characters? Size? Is there any evidence for hybridization from previous phylogenetic studies? It strikes me that an interesting avenue for further thought and/or future studies is discussing how constraints to substrate-borne signaling may drive additional isolating mechanisms (if that is indeed what is going on).

Line 423-425: is there any genetic evidence? Any predictions about whether pre- or post- isolating mechanisms are more likely to evolve? What about chemical signals? Peri-copulation behaviors?

Line 426-437: this would be a great place to discuss how important constraints could be. In the case of this system, signal interference may dominate given signal production constraints, noise, and competition

Line 443-449: Great points! This may be a good place to say that signal interference (possibly driven by constraints) may require a rethinking of soundscape ecology. A different set of rules may govern soundscape ecology in substrate-borne signals.

Line 485: additionally potential tradeoffs between con- and heterospecific interference!

Reviewer #3 (Remarks to the Author):

The authors have undertaken a comprehensive revision of their manuscript, including additional tests of the effect of noise conditions on acoustic measurements. The main results have not changed, but previous conclusions about the spiders' potentially changing the dominant frequency of their signals under different conditions have not held up and have been removed. I think the additional tests are a value-added component of the paper, and will serve as a caveat to other researchers wanting to use these or similar methods.

I have no further changes to suggest to the manuscript. The only minor change I would suggest is to provide a more detailed explanation of the statistical tests presented in Table S2. It took me a while to figure out what was being presented in that table. Another two sentences in the table legend would be sufficient to enable readers to understand the results: an example of one of the statistical models, and a statement indicating what a significant result means.

Author Response: Thank you for the opportunity to revise our manuscript. In the responses below, our responses to reviewer comments can be found in blue.

Important updates:

1. We refined the definition of noise.
2. We added more references as suggested by Reviewer 1.

Figure updated:

1. In figure 2, we entered the number of total detections for each type of vibrations.
2. In figure 3, we changed ‘vibratory sounds’ to ‘substrate-borne vibrations’.

Reviewers' comments:

Reviewer #1:

The revised version of the manuscript is significantly improved. However, there are still some points mentioned below that require further clarification.

1. The authors have added a definition of noise, as suggested. However, this definition (airborne+substrate-borne, abiotic/biotic noise) only refers to what they call "general noise". Four different noise terms are used in the paper: Background noise (line 166), general noise (lines 235, 276), non-biological background noise and local background noise. The latter two terms are used in the Supplement ppt, where non-biological background noise is defined as "noise originating from recording equipment". Since "noise" or "noisiness" play such an important role in this paper, I believe that more detailed information on each term needs to be provided in the text, including the definition of abiotic and biotic noise. For example, two examples of recorded airborne sounds are shown in Fig. 2, namely ‘airplane’ and "unknown noise". The former is referred to in the text as "anthropogenic noise". Should this type of noise also be included in the definition of general noise? If the authors define noise as anything that can interfere with the vibrational communication of spiders, then heterospecific signals are part of general noise. Are the heterospecific signals (Objective III) only *Schizocosa* vibrational signals, all heterospecific vibrational signals, or all communication signals recorded in the substrate, including bird songs (see lines 346-348).

- We revised the definition of ‘general noise’ to ‘all detected airborne and substrate-borne vibrations other that belong to different types’ (Line 250-251; Line 306-307). We removed the term ‘abiotic/biotic noise’ to avoid confusion.
- We replaced the term in the supplementary ppt (‘non-biological background noise’, ‘local background noise’) with ‘background noise’ to avoid confusion.
- We specified that the analysis of the conspecific/heterospecific signals is about *S. stridulans* and *S. uetzi* in the methods section (Line 252; Line 296-298; Line 309-310).

How did the authors count the noise? It would be helpful if they could define abundance and diversity (lines 241-242).

- With the revised definition of general noise, the definition of abundance and diversity is more clear to understand (Line 249-250). Thank you for your suggestions.

In the results, it would be helpful if the authors mentioned which were the most common sources of abiotic and biotic noise in their recordings.

- We entered the total number of detections of each type of vibrations in Figure 2.

2. With the advent of biotremology as a distinct discipline separate from bioacoustics (Hill & Wessel 2016; Hill et al. 2019), the terminology as "vibratory sound" is misleading. It should be either airborne sound or substrate vibrations. Vibrations induced by airborne sound sources in the substrate are still substrate vibrations. Vibroscape is a vibrational, not an acoustic environment (lines 389-390). Check the entire text for consistency.

- We changed the term. Thank you for your suggestion.

3. Line 47: Insert reference Sturm et al. 2021

- Inserted the reference

4. Line 49: Insert reference Sturm et al. 2019

- Inserted the reference

5. Some references used in the text are missing in the reference list (e.g. Gross et al. 2010, Schwartz & Bee 2013, Sturm et al. 2019). Check the entire text.

- We added the references to the reference list. Thank you for pointing out.

6. Lines 142-143: change to... "would be simultaneously picking up a single individual".

- Changed

7. Lines 160-163: if I understand correctly, all 17 "sounds" shown in Fig. 2 were detected, but not necessarily identified by automatic signal detection as well. Were the spider signals classified by visual inspection of the spectrograms (lines 194-195)? Bird songs were identified by BirdNET (lines 195-198). How were other species shown in Fig. 2 classified?

- We classified the insects' sounds by the Library of Singing Insects of North America (SINA; Walker & Moore, 2003) and the anecdotal observations in the field (e.g. airplane sounds). We added the information in the materials and methods section (Line 209-211).

8. Line 325: delete one 'the'.

- Deleted

9. Line 341: 'Duplex' in italics.

- Corrected

10. Line 376: "Unprecedented" is a very strong word. Could it be replaced by another one?

- We changed "unprecedented" to "large".

11. Lines 397-403: I suggest rewording the text and also including the reference Lohrey et al. 2009. It should also be mentioned in the discussion that these two studies (Gordin & Uetz 2012, Lohrey et al. 2009) were conducted in the laboratory on artificial substrates (plastic/paper).

- We included Lohrey et al. 2009 and rewrote the sentences (Line 412-420).
- "... Previous studies suggested that airborne sounds transmitted to the artificial substrates (e.g. filter paper) influence the substrate-borne communication of another ground-dwelling wolf spider in laboratory conditions (*Schizocosa ocreata* – Lohrey et al., 2009; Gordon & Uetz, 2012), but further studies are required to understand whether and how the airborne sounds transmitted to natural substrates (e.g. leaf litter) influence the communication of ground-dwelling arthropods including spiders."

12. Is there any reference to support the hypothesis that arthropods can adjust the spectral properties of their vibrational signals in the short term?

- We could not find proper evidence of plastic alteration of spectral properties (e.g. dominant frequency) in arthropods, but other studies are showing that male wolf spiders can alter their courtship activities and signal structures (e.g. multimodality) responding to short-term environmental changes.

13. Line 470: Insert space between *S. stridulans*.

- Corrected

14. Lines 471-474: What is meant by "...heterospecific signals at the moment of signal production"?

- We revised the sentence.
- Changed to: "In response to how abundant *S. uetzi* signals are around courting *S. stridulans* males, *S. stridulans* males decreased the complexity of their vibratory signals by reducing the duration of idles (Figure 9, Table 3)."

15. Lines 502-515: The importance of a reference library of vibrational signals has been recognized previously, and authors should cite some of the earlier work discussing this issue (e.g. Sturm et al. 2019, 2021, 2022; Frommolt et al. 2019; Akassou et al. 2022).

- We added the references. Thank you for your suggestions.

Reviewer #2:

The authors describe a low cost technique to record and evaluate vibrosapes across plant litter microhabitats. Using automated techniques, the authors demonstrate the technique in identifying wolf spider vibratory displays and use it to test several hypothesis regarding signaling niche use. The authors find separation in the signaling niches of some but not all wolf spider species observed. Additionally, the authors found evidence for plasticity in signaling behavior in response to environmental noise. Overall, this article is excellent. The techniques described will be of great use to bioacousticians and behavioral ecologists. Additionally, the authors show one of the ways that this technique can be used to answer important questions in behavioral ecology in ways that have not been possible before. I had a few reservations however. First, the limitations of the technique was hard to assess particularly on whether relative rarity of vibratory songs had to do with the absence of an extensive vibratory song library or the dominance of wolf spiders in the habitats measured. Justification of some of the analytical parameters

used is also needed (e.g. frequency and temporal bands). Finally, I had a few concerns regarding the interpretations of the niche overlap data. I look forward to seeing this article in print! This will be a major step forward for biologists interested in animal communication.

The authors have improved the manuscript and adequately dealt with my concerns. A few, very minor thoughts and points are detailed below. I look forward to seeing this in print!

1. Line 19: rewrite to "...to acquire in many groups."

- Changed

2. Line 24-26: suggest deleting this sentence. I do not think that the challenges stated in the previous sentence are any worse in modalities that humans cannot detect. Do the authors mean to say that these problems are exacerbated because they are multiplied by detection difficulties?

- We wrote the sentence to stress the importance of studies on vibrosapes because the detection difficulties by human observers may lead researchers to overlook the ecologically important dimensions of soundscapes. We rewrote the sentence to make it clear (Line 36-39).
- Changed to: "These challenges become particularly difficult to overcome when communication occurs in sensory modalities beyond the range of human perception that cannot be directly detected by human observers (e.g., near-field sound or substrate-borne vibrations)."

3. Line 47-50: limited detection range will also be a problem with individual sensors. Suggest rewriting to "limited detection range of a single device...". Suggest emphasizing that the large scale is possible through the use of multiple devices later.

- We rewrote the sentence. Thank you for your suggestion (Line 61-65).
- Changed to: "Unfortunately, the limited detection range of a single device hinders investigations of vibrosapes in the field as it is particularly difficult to conduct studies across large temporal and spatial scales (Štrum et al., 2019; 2021; 2022). While the detection range can be improved by using multiple devices, it currently limits participation in the study of vibrosapes and ecometology to laboratories and investigators with access to such expensive equipment."

4. Line 76: suggest mentioning temporal properties as manuscript includes discussion of this

- We changed the sentence (Line 86-91).
- Changed to: "...and/or diverging signal properties such as spectral ranges or temporal patterns..."

5. Line 86: delete "can"

- Deleted

6. Line 102: sympatric, closely

- Changed

7. Line 106: "variation in signaling behaviors"

- Corrected

8. Line 126: mature

- Corrected

9. Line 142: rewrite to “to travel (Uetz et al., 2013) thus reducing the possibility that...”

- Changed

10. Line 177: is plotting the changes in noise level informative? It may be interesting to show changes in overall “noise” or amplitude levels across a day particularly as things such as wind noise could be an important constraint on signaling behavior

- It would be interesting to know if the amplitude of background noise varies across recording periods, but the background noise level in our recording was affected by many non-environmental factors (e.g. internal noise from audio cables/devices). As we discussed in the discussion, the improvement of recording equipment will enable a more detailed investigation of the effects of many ecological and environmental factors on vibrosapes.

11. Line 259-262: I am confused by this sentence. Would noise filtering distort these other measurements?

- Yes. According to further verification of our adaptive filtering method, the other measurements can be distorted by the variation in the background noise profile (Supplementary Materials S1). We revised the sentence to avoid the confusion (Line 507-521).
- Changed to: “We chose these two characters due to their relative robustness to our noise filtering method which may distort the measurement of other acoustic characters depending on the background noise profiles (e.g. frequency range, signal-to-noise ratio; Supplementary Materials S1).”

12. Line 312: what is the cricket species? List if known

- It was Jamaican field cricket (*Gryllus assimilis*). We identified the song using the Library of Singing Insects of North America (SINA).

13. Line 341: duplex not italicized

- Corrected

14. Line 349-368: does this suggest different forms of plasticity are at play in regards to hetero and conspecific interference? This is very interesting to think about.

- Yes. The different responses to hetero- and conspecific interference may be explained by multiple hypotheses. For instance, the abundance of conspecific signals may reduce the general signal activities in the context of eavesdropping and/or male-male competition, and the abundance of heterospecific signals may increase the risk of signal interference and potential hybridization. We introduced the hypotheses in the discussion (Line 471-505).

15. Line 378: But aren't these known because they have been identified as Schizocosa species? Suggest rewriting

- We rewrote the sentence. Thank you for pointing out (Line 393-395).
- Changed to: “...we were able to identify 10 airborne sounds and 4 substrate-borne sounds with 3 unknown substrate-borne sounds including courtship songs of three species of Schizocosa wolf spider.”

16. Line 415: Are these species diverging in morphological characters? Size? Is there any evidence for hybridization from previous phylogenetic studies? It strikes me that an interesting avenue for further thought and/or future studies is discussing how constraints to substrate-borne signaling may drive additional isolating mechanisms (if that is indeed what is going on).

- The cited studies used passive acoustic monitoring (PAM) methods, so morphological information is not available.
- Although there are a few studies on hybridization in *Schizocosa* wolf spiders (e.g. Stratton & Uetz, 1986), it has not yet been tested whether and how the potential risk of reproductive interference promotes the evolution of their multimodal courtship signals. As we suggested in the discussion (Line 498-501), the hypothesis should be validated by controlled laboratory experiments such as playback experiments.

17. Line 423-425: is there any genetic evidence? Any predictions about whether pre- or post- isolating mechanisms are more likely to evolve? What about chemical signals? Peri-copulation behaviors?

- Currently, we do not have any genetic evidence that supports the potential hybridization between *S. stridulans* and *S. uetzi*. To validate the hypothesis, various mechanisms for species isolation should also be tested. We wrote about the limited evidence in the discussion (Line 442-444).

18. Line 426-437: this would be a great place to discuss how important constraints could be. In the case of this system, signal interference may dominate given signal production constraints, noise, and competition

- We discussed how the constraints of substrate-borne vibratory communication may influence the evolution of time, space, and spectral characteristics of vibratory signals (Line 457-468). Thank you for your suggestions.

19. Line 443-449: Great points! This may be a good place to say that signal interference (possibly driven by constraints) may require a rethinking of soundscape ecology. A different set of rules may govern soundscape ecology in substrate-borne signals.

- Thank you for your suggestions. We discussed the hypothesis that the evolutionary constraints may evolve behavioral plasticity in male signaling behaviors of *Schizocosa* wolf spiders, but our results may not be enough to draw a general conclusion about overall vibrosapes consisting of multiple known/unknown ground-dwelling animals.

20. Line 485: additionally potential tradeoffs between con- and heterospecific interference!

- Yes. We revised the sentence (Line 502-505).
- Changed to: "... this potential trade-off between accurate species recognition/detection and preference for complexity of conspecific females would be an interesting subject of study to understand how animals evolve complex communication displays."

Reviewer #3:

The authors have undertaken a comprehensive revision of their manuscript, including additional tests of the effect of noise conditions on acoustic measurements. The main results have not changed, but previous conclusions about the spiders' potentially changing the dominant frequency of their signals under different conditions have not held up and have been removed. I think the additional tests are a value-added

component of the paper, and will serve as a caveat to other researchers wanting to use these or similar methods.

I have no further changes to suggest to the manuscript. The only minor change I would suggest is to provide a more detailed explanation of the statistical tests presented in Table S2. It took me a while to figure out what was being presented in that table. Another two sentences in the table legend would be sufficient to enable readers to understand the results: an example of one of the statistical models, and a statement indicating what a significant result means.

- Thank you for your thorough review and helpful comments. We revised the table legend of Table S2 for better understanding.

Reviewers' comments:

Reviewer #1 (Remarks to the Author):

The revised version of the manuscript is significantly improved. However, there are still some points mentioned below that require further clarification.

1. The authors have added a definition of noise, as suggested. However, this definition (airborne+substrate-borne, abiotic/biotic noise) only refers to what they call "general noise". Four different noise terms are used in the paper: Background noise (line 166), general noise (lines 235, 276), non-biological background noise and local background noise. The latter two terms are used in the Supplement ppt, where non-biological background noise is defined as "noise originating from recording equipment". Since "noise" or "noisiness" play such an important role in this paper, I believe that more detailed information on each term needs to be provided in the text, including the definition of abiotic and biotic noise. For example, two examples of recorded airborne sounds are shown in Fig. 2, namely 'airplane' and "unknown noise". The former is referred to in the text as "anthropogenic noise". Should this type of noise also be included in the definition of general noise? If the authors define noise as anything that can interfere with the vibrational communication of spiders, then heterospecific signals are part of general noise. Are the heterospecific signals (Objective III) only *Schizocosa* vibrational signals, all heterospecific vibrational signals, or all communication signals recorded in the substrate, including bird songs (see lines 346-348). How did the authors count the noise? It would be helpful if they could define abundance and diversity (lines 241-242). In the results, it would be helpful if the authors mentioned which were the most common sources of abiotic and biotic noise in their recordings.

2. With the advent of biotremology as a distinct discipline separate from bioacoustics (Hill & Wessel 2016; Hill et al. 2019), the terminology as "vibratory sound" is misleading. It should be either airborne sound or substrate vibrations. Vibrations induced by airborne sound sources in the substrate are still substrate vibrations. Vibroscape is a vibrational, not an acoustic environment (lines 389-390). Check the entire text for consistency.

3. Line 47: Insert reference Sturm et al. 2021

4. Line 49: Insert reference Sturm et al. 2019

5. Some references used in the text are missing in the reference list (e.g. Gross et al. 2010, Schwartz & Bee 2013, Sturm et al. 2019). Check the entire text.

6. Lines 142-143: change to... "would be simultaneously picking up a single individual".

7. Lines 160-163: if I understand correctly, all 17 "sounds" shown in Fig. 2 were detected, but not necessarily identified by automatic signal detection as well. Were the spider signals classified by visual inspection of the spectrograms (lines 194-195)? Bird songs were identified by BirdNET (lines 195-198). How were other species shown in Fig. 2 classified?

8. Line 325: delete one 'the'.

9. Line 341: 'Duplex' in italics.

10. Line 376: "Unprecedented" is a very strong word. Could it be replaced by another one?

11. Lines 397-403: I suggest rewording the text and also including the reference Lohrey et al. 2009. It should also be mentioned in the discussion that these two studies (Gordin & Uetz 2012, Lohrey et al.

2009) were conducted in the laboratory on artificial substrates (plastic/paper).

12. Is there any reference to support the hypothesis that arthropods can adjust the spectral properties of their vibrational signals in the short term?

13. Line 470: Insert space between *S. stridulans*.

14. Lines 471-474: What is meant by "...heterospecific signals at the moment of signal production"?

15. Lines 502-515: The importance of a reference library of vibrational signals has been recognized previously, and authors should cite some of the earlier work discussing this issue (e.g. Sturm et al. 2019, 2021, 2022; Frommolt et al. 2019; Akassou et al. 2022).

Reviewer #2 (Remarks to the Author):

Title: Vibroscape analysis reveals acoustic niche overlap and plastic alteration of vibratory courtship signals in ground-dwelling wolf spiders

The authors have improved the manuscript and adequately dealt with my concerns. A few, very minor thoughts and points are detailed below. I look forward to seeing this in print!

Line 19: rewrite to "...to acquire in many groups."

Line 24-26: suggest deleting this sentence. I do not think that the challenges stated in the previous sentence are any worse in modalities that humans cannot detect. Do the authors mean to say that these problems are exacerbated because they are multiplied by detection difficulties?

Line 47-50: limited detection range will also be a problem with individual sensors. Suggest rewriting to "limited detection range of a single device...". Suggest emphasizing that the large scale is possible through the use of multiple devices later.

Line 76: suggest mentioning temporal properties as manuscript includes discussion of this

Line 86: delete "can"

Line 102: sympatric, closely

Line 106: "variation in signaling behaviors"

Line 126: mature

Line 142: rewrite to "to travel (Uetz et al., 2013) thus reducing the possibility that..."

Line 177: is plotting the changes in noise level informative? It may be interesting to show changes in overall "noise" or amplitude levels across a day particularly as things such as wind noise could be an important constraint on signaling behavior

Line 259-262: I am confused by this sentence. Would noise filtering distort these other measurements?

Line 312: what is the cricket species? List if known

Line 341: duplex not italicized

Line 349-368: does this suggest different forms of plasticity are at play in regards to hetero and conspecific interference? This is very interesting to think about.

Line 378: But aren't these known because they have been identified as Schizocosa species? Suggest rewriting

Line 415: Are these species diverging in morphological characters? Size? Is there any evidence for hybridization from previous phylogenetic studies? It strikes me that an interesting avenue for further thought and/or future studies is discussing how constraints to substrate-borne signaling may drive additional isolating mechanisms (if that is indeed what is going on).

Line 423-425: is there any genetic evidence? Any predictions about whether pre- or post- isolating mechanisms are more likely to evolve? What about chemical signals? Peri-copulation behaviors?

Line 426-437: this would be a great place to discuss how important constraints could be. In the case of this system, signal interference may dominate given signal production constraints, noise, and competition

Line 443-449: Great points! This may be a good place to say that signal interference (possibly driven by constraints) may require a rethinking of soundscape ecology. A different set of rules may govern soundscape ecology in substrate-borne signals.

Line 485: additionally potential tradeoffs between con- and heterospecific interference!

Reviewer #3 (Remarks to the Author):

The authors have undertaken a comprehensive revision of their manuscript, including additional tests of the effect of noise conditions on acoustic measurements. The main results have not changed, but previous conclusions about the spiders' potentially changing the dominant frequency of their signals under different conditions have not held up and have been removed. I think the additional tests are a value-added component of the paper, and will serve as a caveat to other researchers wanting to use these or similar methods.

I have no further changes to suggest to the manuscript. The only minor change I would suggest is to provide a more detailed explanation of the statistical tests presented in Table S2. It took me a while to figure out what was being presented in that table. Another two sentences in the table legend would be sufficient to enable readers to understand the results: an example of one of the statistical models, and a statement indicating what a significant result means.

Author Response: Thank you for the opportunity to revise our manuscript. In the responses below, our responses to reviewer comments can be found in blue.

Important updates:

Figure updated:

Reviewers' comments:

Reviewer #1:

I have no additional comments and I recommend the paper for publication.

- Thank you very much for your helpful comments.